# You Only Prune Once: A Zero-Shot, Data-Free Pruning at Initialization via Low-Rank Residual Saliency

## Abstract

Pruning at initialization (PaI) seeks sparse subnetworks that can be trained from scratch without iterative retraining or post-hoc compression. Most existing PaI methods rely on data, gradients, or iterative structural optimization, and their saliency scores are typically coupled to a specific sparsity budget. This work introduces a zero-shot, data and gradient-free pruning criterion based on nonnegative low-rank residual saliency. At random initialization, a once-only ordering of parameters is obtained by measuring their deviation from a low-rank additive template in the absolute weight space. This fixed ordering can be re-thresholded to realize arbitrary sparsity levels without rescoring, decoupling parameter ranking from sparsity budget and dataset.

Structural and dynamical analyses provide insight into the effectiveness of residual-based pruning. Spectral evaluation shows stronger post-pruning low-rank concentration than competing methods, while neural tangent kernel diagnostics indicate alignment between residual magnitude and functional influence. Empirical results across CIFAR-10/100, Tiny-ImageNet, ImageNet, and modern ConvNeXt architectures demonstrate competitive or superior performance relative to gradient-based and topology-driven PaI baselines, particularly at extreme sparsity ($\geq 99\%$), alongside substantial reductions in pruning time. These findings suggest that a once-only, dataset-agnostic saliency ordering can reliably identify trainable sparse subnetworks from intrinsic structural properties of random initialization.[1]

## 1 Introduction

Modern deep networks are heavily overparameterized, yet surprisingly structured at random initialization. This structural redundancy motivates *pruning at initialization* (PaI): identifying sparse subnetworks *before* training that can be optimized from scratch under a standard recipe. The Lottery Ticket Hypothesis (LTH) formalized this possibility by showing that dense random networks contain sparse "winning tickets" that match dense performance when trained in isolation (Frankle & Carbin, 2019). Since then, a large body of work has sought practical criteria for discovering such subnetworks directly at initialization (Frankle et al., 2020; Evci et al., 2020; Ramanujan et al., 2020; Natale et al., 2024), and explored dynamic sparse training variants (Kusupati et al., 2020; Jayakumar et al., 2020; Savarese et al., 2020).

Most existing PaI methods score parameters using data, gradients, or higher-order derivatives (e.g., SNIP (Lee et al., 2019), GraSP (Wang et al., 2020)), or enforce iterative conservation laws to avoid layer collapse (e.g., SynFlow (Tanaka et al., 2020)). More recent methods emphasize topology and connectivity. Node–path balancing principles explicitly optimize structural metrics to stabilize extreme sparsity (Pham et al., 2023; Xiang et al., 2025), while expander-style constructions impose graph-theoretic connectivity priors (Stewart et al., 2023; Prabhu et al., 2018; Hoory et al., 2006). Connectivity-focused analyses demonstrate that effective sparsity and path preservation critically influence trainability (Vysogorets & Kempe, 2023). Restricted random pruning at initialization (Otsuka et al., 2024) and structural pruning in graph transformers (Ito et al., 2025) further highlight that strong lottery tickets can emerge from architectural structure alone.

---

[1] Anonymous implementation of YOPO is available at `https://anonymous.4open.science/status/YOPO-7F95`.

Broader surveys summarize the growing taxonomy of pruning strategies and their trade-offs (Wang et al., 2022; Cheng et al., 2024); a comprehensive discussion of related work is provided in Appendix C.

**Unstructured Pruning: Limitations and Benefits.** Like most PaI baselines, our approach performs *unstructured* pruning – zeroing individual scalar weights rather than entire filters or channels. While the resulting subnetwork has a proportional theoretical reduction in multiply–accumulate (MAC) operations, translating this into wall-clock speedup on modern hardware (GPUs, TPUs) is non-trivial without specialized sparse-kernel support. Dense computation units process all memory addresses regardless of sparsity, so zero entries contribute minimally to latency reduction in standard frameworks. This limitation is common to all unstructured PaI methods (Lee et al., 2019; Wang et al., 2020; Tanaka et al., 2020), and is discussed further in Section D.

Despite this practical constraint, unstructured pruning offers several advantages that motivate its continued study across the field. It provides the finest granularity of parameter selection, allowing saliency criteria to operate at the individual weight level without being constrained by coarse structural units such as filters or channels. This flexibility is particularly valuable at initialization, where the importance of specific parameters is inherently uncertain. Unstructured sparsity also yields direct reductions in model size and memory footprint, which can be significant at high compression ratios even in the absence of proportional runtime gains. Furthermore, empirical studies have shown that unstructured masks can exhibit emergent coarse-grained patterns – for instance, entire filters or channel connections may be implicitly eliminated as a consequence of fine-grained saliency, providing a potential bridge toward structured compression (Cheng et al., 2024). These considerations suggest that unstructured PaI remains a valuable tool for studying the sparse trainability of neural networks, with structured acceleration as a natural downstream step.

Beyond hardware-alignment considerations, existing PaI algorithms also face two fundamental algorithmic limitations. First, most criteria are *budget-coupled*: changing sparsity typically requires recomputation or iterative refinement. Second, saliency measures are often implicitly *dataset- or seed-dependent*, limiting mask reuse and cross-task transferability. Recent theoretical results suggest fundamental barriers to arbitrary pruning at initialization (Kumar et al., 2024), indicating that effective criteria must exploit intrinsic structural regularities already present in random weights.

These limitations point to a deeper structural question:

*Does there exist a once-only, dataset-agnostic ordering of parameters at random initialization that induces competitive sparse subnetworks across sparsity budgets and datasets?*

We argue that randomly initialized convolutional weight tensors contain a pronounced *low-rank additive template structure*. Spectral analyses of random networks and training dynamics (Pennington et al., 2017; Jacot et al., 2018) indicate the presence of dominant shared modes even prior to learning. Parameters aligned with these modes are structurally redundant, whereas parameters that deviate from them encode distinctive routing patterns across channels and paths.

Based on this perspective, this work introduces You Only Prune Once (YOPO)[2], a zero-shot, data and gradient-free PaI method that constructs a saliency ordering from the *nonnegative low-rank residual* of absolute weights. For a prunable layer with $\tilde{W} = |W|$, we compute a nonnegative matrix factorization $\tilde{W} \approx VH$ and define saliency as $S = |\tilde{W} - VH|$. Because this residual is computed once at initialization, it induces a fixed parameter ranking that can be re-thresholded to realize arbitrary sparsity budgets without rescoring.

Beyond proposing a masking rule, we provide structural and dynamical evidence explaining why low-rank residual pruning is effective. Spectral analysis reveals that YOPO induces the steepest singular value decay among competing pruning strategies, concentrating surviving parameters into a lower-dimensional subspace and yielding the strongest low-rank compressibility. This supports the *low-rank template hypothesis*: initialization weights contain dominant shared modes, and pruning their residual isolates structurally distinctive connections. At extreme sparsity levels, YOPO implicitly achieves a balanced regime between node-dominant

---

[2]Once-only refers to computing the saliency ranking a single time at random initialization; subsequent sparsity levels are obtained via monotone re-thresholding without recomputation.

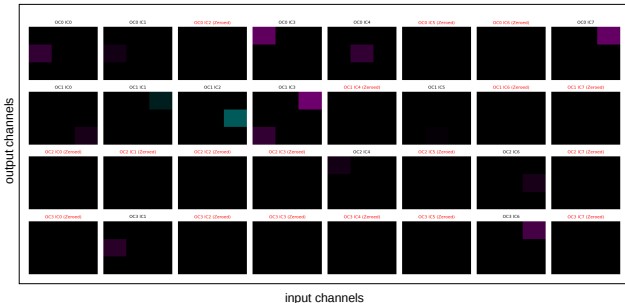 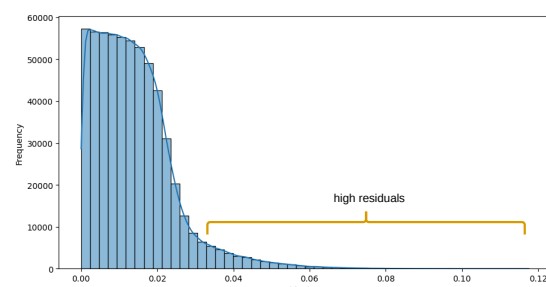

**(a) Kernel-level sparsity.** Each cell denotes a 2D kernel between input–output channels. Fully black squares indicate completely removed channel connections. Cyan and magenta represent negative and positive weights with intensity proportional to magnitude.

**(b) NMF residual distribution.** Histogram of residual saliency scores. The heavy-tailed shape separates low-residual (redundant) from high-residual (structurally distinctive) parameters.

**Figure 1: Kernel sparsity and residual saliency under YOPO.** (a) Visualization of convolutional kernels after pruning. Entirely black squares denote removed channel connections, revealing structured kernel-level sparsity, while partially pruned kernels show fine-grained element-wise removal. (b) Residual saliency exhibits a skewed, heavy-tailed distribution, with pruning concentrated on low-residual weights and retention of the high-residual tail.

and path-dominant topologies. In contrast to explicit node–path balancing methods that optimize discrete structural objectives (Pham et al., 2023; Xiang et al., 2025), YOPO approximates this balance without solving combinatorial or differentiable programs, yielding superior generalization at sparsities above 99%. Our work further shows that residual magnitude correlates with parameter influence measured through the diagonal of the neural tangent kernel (NTK) (Jacot et al., 2018), with median Spearman correlation around 0.41 across convolutional layers. Thus, parameters that deviate most from the low-rank template tend to be functionally influential in training dynamics. Moreover, YOPO preserves NTK trace at levels comparable to intelligent pruning methods while significantly reducing computational cost. Together, these results suggest that low-rank residual pruning couples *structural redundancy detection* with *functional influence alignment*, providing a principled mechanism for initialization-time sparsification.

Empirically, across CIFAR-10/100, Tiny-ImageNet, ImageNet, and modern ConvNeXt architectures, YOPO matches or surpasses strong gradient-based and topology-driven PaI baselines (Lee et al., 2019; Wang et al., 2020; Tanaka et al., 2020; Pham et al., 2023; Xiang et al., 2025; Otsuka et al., 2024). At the same time, it reduces pruning time by one to two orders of magnitude and uniquely supports mask reuse across sparsity budgets and datasets without recomputation.

**Our Contribution.** This work introduces You Only Prune Once (YOPO), a zero-shot, data and gradient-free pruning-at-initialization framework based on nonnegative low-rank residual saliency. The method constructs a once-only parameter ordering at random initialization and decouples saliency from sparsity budget via monotone re-thresholding, enabling exact global or layer-wise sparsity without rescoring and supporting mask reuse across datasets. Structural and dynamical analyses provide justification for residual-based pruning, including stronger post-pruning spectral concentration and consistent alignment between residual magnitude and NTK-based influence measures. Extensive experiments across CIFAR-10/100, Tiny-ImageNet, ImageNet, and modern ConvNeXt architectures demonstrate robustness at extreme sparsity levels while significantly reducing pruning time relative to optimization- or gradient-driven PaI methods.

## 2    Structural Properties of Random Initialization

Begin with examining the intrinsic geometric structure of randomly initialized convolutional layers. Although initialization schemes are designed to preserve variance across layers, the resulting weight tensors exhibit nontrivial spectral regularities when viewed as matrices. These regularities motivate a structural interpretation of redundancy at initialization.

**Low-Rank Template Hypothesis.** Consider a convolutional layer with weight tensor $W \in \mathbb{R}^{o \times i \times k_h \times k_w}$. Reshaping along the channel dimension yields $\tilde{W} \in \mathbb{R}^{o \times d}$, where $d = ik_h k_w$. Let the singular value decomposition (SVD) of $\tilde{W}$ be

$$\tilde{W} = U \Sigma V^\top, \tag{1}$$

where $\Sigma = \mathrm{diag}(\sigma_1, \ldots, \sigma_r)$ with $\sigma_1 \geq \cdots \geq \sigma_r \geq 0$ and $r = \mathrm{rank}(\tilde{W})$.

To measure spectral concentration, define the cumulative explained variance ratio

$$\mathrm{EVR}(k) = \frac{\sum_{j=1}^{k} \sigma_j^2}{\sum_{j=1}^{r} \sigma_j^2}. \tag{2}$$

Empirically, $\mathrm{EVR}(k)$ grows rapidly for moderate $k$, indicating that a relatively small number of singular directions accounts for a substantial portion of the Frobenius energy $\|\tilde{W}\|_F^2 = \sum_{j=1}^{r} \sigma_j^2$. Thus, even at random initialization, convolutional weight matrices admit an effective rank significantly smaller than their full dimension.

This observation motivates the *low-rank template hypothesis*: random convolutional weights contain dominant additive modes shared across output channels, which act as global structural templates. Higher-order components represent deviations from these dominant modes.

Formally, for $k \ll r$, the truncated approximation

$$\tilde{W}_k = \sum_{j=1}^{k} \sigma_j u_j v_j^\top \tag{3}$$

captures the principal shared structure, while $\tilde{W} - \tilde{W}_k$ contains residual variation not explained by the dominant modes.

**Additive Structure and Nonnegativity.** While SVD reveals compressibility, its components mix positive and negative contributions and allow cancellation across modes. For initialization-time sparsification, such cancellation obscures the interpretation of shared additive structure.

To isolate purely additive structure, we instead consider the elementwise absolute weight matrix $\tilde{W} = |W|$, and approximate it via nonnegative matrix factorization (NMF),

$$\tilde{W} \approx VH, \qquad V \in \mathbb{R}_{\geq 0}^{o \times k}, \quad H \in \mathbb{R}_{\geq 0}^{k \times d}. \tag{4}$$

Under this model, each output channel is expressed as a nonnegative combination of $k$ basis templates. Unlike SVD, the decomposition in equation 4 is strictly additive: $VH$ captures shared activation patterns without sign cancellation. The nonnegativity constraint enforces a parts-based representation, which is particularly natural for convolutional filters, where magnitude encodes contribution strength. The approximation $VH$ can therefore be interpreted as the dominant additive template underlying the layer, while deviations from this template encode structurally distinctive behavior.

**Structural Redundancy and Distinctiveness.** Define the residual matrix

$$R = \tilde{W} - VH. \tag{5}$$

For each parameter $(i, j)$, the magnitude $|R_{ij}|$ measures the extent to which $\tilde{W}_{ij}$ deviates from the dominant additive template. This yields a structural characterization at initialization. A parameter is *template-aligned* if $|R_{ij}|$ is small, meaning it lies near the low-dimensional manifold spanned by the columns of $H$. Such parameters are structurally redundant in the sense that their contribution can be reconstructed from shared modes. Conversely, parameters with large residual are poorly approximated by the template and represent structurally distinctive deviations.

Crucially, this notion of redundancy is entirely intrinsic to the weight geometry and requires no data, gradients, or task information. It depends solely on the internal organization of $\tilde{W}$ at random initialization.

This structural viewpoint reframes initialization-time sparsification as the problem of identifying parameters that deviate most strongly from dominant additive templates. In the following section, we formalize this principle as a once-only residual-based saliency ordering.

## 3 Once-Only Low-Rank Residual Pruning (YOPO)

We now formalize a residual-based sparsification principle derived from the structural model in Section 2. The construction is performed once at random initialization and induces a fixed ordering of parameters that supports arbitrary sparsity budgets via re-thresholding.

**Nonnegative Low-Rank Factorization.** Let $\mathcal{L}$ denote the set of prunable layers of a fixed architecture. For each convolutional layer $\ell \in \mathcal{L}$ with weight tensor $W^{(\ell)} \in \mathbb{R}^{o_\ell \times i_\ell \times k_\ell^h \times k_\ell^w}$, define the flattened absolute matrix

$$\tilde{W}^{(\ell)} = \left|\text{flat}(W^{(\ell)})\right| \in \mathbb{R}_{\geq 0}^{o_\ell \times d_\ell}, \qquad d_\ell = i_\ell k_\ell^h k_\ell^w.$$

For a prescribed rank $r_\ell \ll \min(o_\ell, d_\ell)$, we compute a nonnegative factorization

$$\tilde{W}^{(\ell)} \approx V^{(\ell)} H^{(\ell)}, \qquad V^{(\ell)} \in \mathbb{R}_{\geq 0}^{o_\ell \times r_\ell}, \quad H^{(\ell)} \in \mathbb{R}_{\geq 0}^{r_\ell \times d_\ell}, \tag{6}$$

as discussed in Section 2. This decomposition extracts dominant additive templates shared across output channels.

**Residual-Based Saliency.** Given the factorization in equation 6, we define the elementwise residual

$$R^{(\ell)} = \tilde{W}^{(\ell)} - V^{(\ell)} H^{(\ell)}, \tag{7}$$

and the corresponding saliency

$$S^{(\ell)} = \left|R^{(\ell)}\right|. \tag{8}$$

The family $S(\theta_0) = \{S^{(\ell)}\}_{\ell \in \mathcal{L}}$ is computed once at initialization. This induces a total ordering of all prunable parameters, independent of data, gradients, or sparsity budgets.

Masks are constructed by thresholding:

$$M^{(\ell)}(\tau) = \mathbf{1}\left[S^{(\ell)} > \tau\right]. \tag{9}$$

The induced sparsity pattern is illustrated in Figure 1, which shows structured kernel-level disconnections and a heavy-tailed residual landscape consistent with selective retention of structurally distinctive parameters.

**Structural Properties of the Saliency Ordering.** The residual-based ordering admits several structural invariances.

**Proposition 1** (Magnitude and perfect-template limits)**.** *Let $S^{(\ell)}$ be defined by equation 8.*

- (a) *(*Rank-0 reduction*) If $r_\ell = 0$ (i.e., $VH \equiv 0$), then $S^{(\ell)} = \tilde{W}^{(\ell)}$; YOPO reduces to magnitude ranking at initialization.*

- (b) *(*Exact template*) If $\text{rank}_+(\tilde{W}^{(\ell)}) \leq r_\ell$ and $VH = \tilde{W}^{(\ell)}$ exactly, then $S^{(\ell)} \equiv 0$. In practice this is a measure-zero event; we prevent degenerate collapse by (i) choosing small $r_\ell$, (ii) using strict thresholds, and (iii) enforcing per-row minimum keep.*

**Lemma 1** (Positive homogeneity and scale stability)**.** *For any scalar $c > 0$, replacing $\tilde{W}^{(\ell)}$ by $c\tilde{W}^{(\ell)}$ admits optimal factors $(\sqrt{c}V^\star, \sqrt{c}H^\star)$ whenever $(V^\star, H^\star)$ is optimal for $\tilde{W}^{(\ell)}$, and the corresponding residual scales as $S^{(\ell)} \mapsto c\,S^{(\ell)}$. Hence the* ranking *of entries in $S^{(\ell)}$ is invariant to positive scalar rescaling of a layer.*

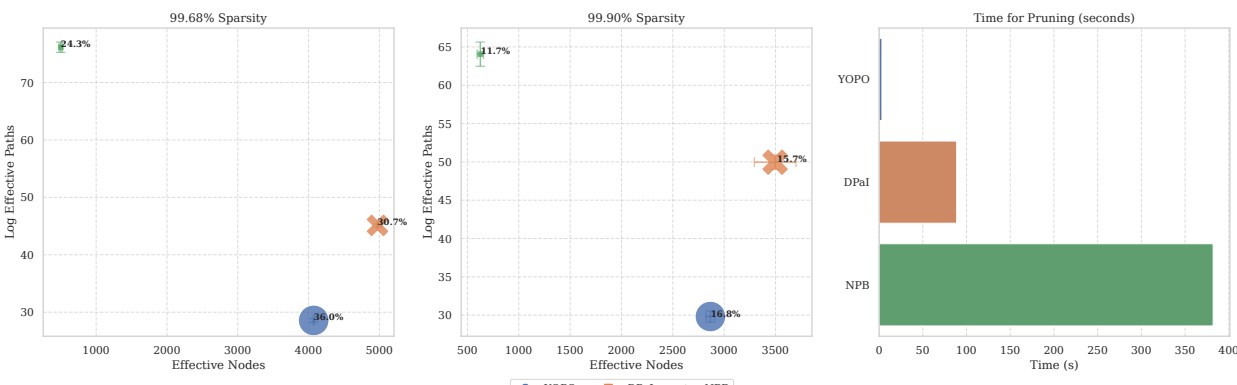

**Figure 2:** Extreme sparsity analysis comparing structural properties and pruning time. **Left:** At 99.68% sparsity, YOPO achieves the highest test accuracy (35.97%) while maintaining the lowest log effective paths, indicating superior structural efficiency. **Middle:** At 99.90% sparsity, YOPO remains the most resilient, preserving the best accuracy (16.75%) under extreme compression. **Right:** YOPO is significantly faster (3.29s) than DPaI and NPB, highlighting its practical efficiency. Overall, YOPO demonstrates a balanced advantage across accuracy, structural compactness, and pruning speed at extreme sparsity levels.

**Lemma 2** (Permutation invariance within flattened structure)**.** *Let $\Pi_r, \Pi_c$ be permutation matrices acting on rows/columns of $\tilde{W}^{(\ell)}$ (e.g., re-indexing output channels or kernel coordinates under flattening). Then $S^{(\ell)}$ permutes accordingly: if $\tilde{W}' = \Pi_r \tilde{W}^{(\ell)} \Pi_c$, an NMF of $\tilde{W}'$ yields $S' = \Pi_r S^{(\ell)} \Pi_c$. Thus once-only rankings respect reindexings inherent to convolutional reshaping.*

**Proposition 2** (Nested masks across budgets)**.** *Let $M^{(\ell)}(\tau) = \mathbf{1}[S^{(\ell)} > \tau]$. If $\tau_1 < \tau_2$, then $M^{(\ell)}(\tau_2) \leq M^{(\ell)}(\tau_1)$ elementwise. Consequently, the sets of survivors are nested as sparsity increases, and all budgets $p \in [0, 1]$ are obtained by* re-thresholding *a fixed $S^{(\ell)}$ without recomputation.*

**Collapse Avoidance via Nonnegativity.** Nonnegativity of the factorization plays a structural role. Because $V^{(\ell)}H^{(\ell)}$ is additive and nonnegative, the residual in equation 7 cannot arise from cancellation between positive and negative modes. Large residual entries therefore correspond to genuine mismatch from dominant additive templates.

To prevent row or channel collapse at extreme sparsity, we enforce minimal survival constraints:

$$\sum_{j=1}^{d_\ell} M_{ij}^{(\ell)} \geq m_{\min}, \qquad \sum_{i=1}^{o_\ell} M_{ij}^{(\ell)} \geq c_{\min}, \tag{10}$$

for small constants $m_{\min}, c_{\min}$.[3]

The additive structure of equation 6 ensures that each output channel contributes to at least one template component, and thus residual mass is typically distributed across channels. In practice, this significantly reduces the risk of neuron or layer collapse without iterative rebalancing procedures. Together, these properties establish that residual-based saliency is once-only, scale-stable, permutation-equivariant, budget-decoupled, and structurally robust.

**Calibration and Exact Budget Control.** The residual-based saliency $S^{(\ell)}$ defined in equation 8 induces a total ordering of parameters within each layer. To obtain a binary mask at a desired sparsity level, we convert this ordering into thresholds.

---

[3]Formal collapse-avoidance guarantees based on nonnegativity and robust thresholding are provided in Appendix A, with detailed proofs therein.

For each layer $\ell$, we define a threshold of the form

$$\tau_\ell(\alpha) = \mathrm{median}(S^{(\ell)}) + \alpha \cdot \mathrm{MAD}(S^{(\ell)}), \tag{11}$$

where $\alpha \geq 0$ controls pruning aggressiveness and MAD denotes the median absolute deviation. An alternative formulation replaces MAD with the standard deviation; both induce monotone threshold families. The mask is then defined as

$$M^{(\ell)}(\alpha) = \mathbf{1}\Big[ S^{(\ell)} > \tau_\ell(\alpha) \Big]. \tag{12}$$

Let the achieved global sparsity be

$$\hat{s}(\alpha) = 1 - \frac{\sum_\ell \| M^{(\ell)}(\alpha) \|_0}{\sum_\ell o_\ell d_\ell}. \tag{13}$$

By the nested mask property established in Proposition 2, $\hat{s}(\alpha)$ is monotone non-decreasing in $\alpha$. Therefore, for any feasible target sparsity $p \in [0, 1]$, there exists $\alpha(p)$ such that $\hat{s}(\alpha(p)) \approx p$. A one-dimensional bisection procedure recovers $\alpha(p)$ to arbitrary tolerance, yielding exact budget control. [4]

Two calibration regimes arise naturally. In global calibration, a single parameter $\alpha$ is shared across layers, allowing sparsity to be allocated adaptively according to residual statistics. In layerwise calibration, each layer receives its own parameter $\alpha_\ell$, enabling fixed per-layer sparsity ratios. Both regimes operate on the same once-only saliency and differ only in how thresholds are selected; no rescoring is required when the sparsity budget changes.

To prevent degenerate collapse under extreme compression, we enforce minimal survival constraints

$$\sum_{j=1}^{d_\ell} M_{ij}^{(\ell)} \geq m_{\min}, \qquad \sum_{i=1}^{o_\ell} M_{ij}^{(\ell)} \geq c_{\min}, \tag{14}$$

for small constants $m_{\min}, c_{\min}$. These constraints are compatible with the nested threshold structure and preserve monotonicity of $\hat{s}(\alpha)$. [5]

## 4 Structural and Dynamical Analysis

We now analyze the structural and functional properties of subnetworks induced by residual-based pruning. The goal is to understand what is preserved-and what is removed-when sparsification is driven by deviation from low-rank additive templates.

**Spectral Concentration and Effective Rank.** We first examine the spectral structure of pruned convolutional layers. Let $\tilde{W}_{\mathrm{dense}}^{(\ell)}$ denote the flattened absolute weight matrix of a dense layer and $\tilde{W}_{\mathrm{mask}}^{(\ell)}$ the corresponding matrix after pruning. Let $\tilde{W}_{\mathrm{mask}}^{(\ell)} = U\Sigma V^\top$ be its singular value decomposition with singular values $\{\sigma_j\}_{j=1}^r$. To quantify concentration, we consider the cumulative explained variance ratio using equation 2. Across representative convolutional layers at $\sim 95\%$ sparsity, YOPO-pruned subnetworks exhibit markedly steeper singular value decay than random and magnitude-based pruning at matched sparsity. In particular, the number of singular components required to reach $\mathrm{EVR}(k) \geq 0.9$ is consistently smaller under YOPO, indicating a reduced effective rank. This demonstrates that residual-based pruning does not merely remove parameters uniformly; rather, it concentrates the remaining energy into a lower-dimensional subspace. This spectral concentration aligns with the heavy-tailed residual distribution observed in Figure 1, where only a small fraction of structurally distinctive weights populate the high-residual tail. The resulting subnetworks are therefore more spectrally compressible than their random or gradient-agnostic counterparts, providing post-pruning evidence for the low-rank template hypothesis introduced in Section 2. The full spectral behavior is visualized in Figure 3(a,b), where YOPO exhibits the steepest singular value decay and the fastest variance saturation among pruning strategies.

---

[4]Monotone re-thresholding (MORT, Alg. 2) enables exact sparsity control without recomputing saliency.
[5]The complete YOPO pruning procedure is outlined in Algorithm 1.

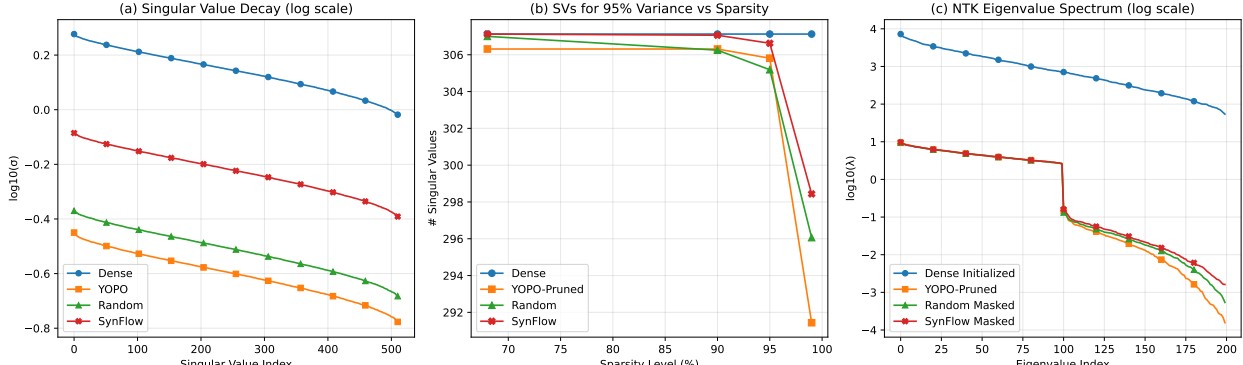

**Figure 3:** Spectral and NTK-level analysis of YOPO. (a) YOPO exhibits the steepest singular value decay, indicating stronger low-rank concentration. (b) Across sparsity levels, YOPO requires fewer singular values to explain 95% of variance. (c) NTK eigenvalue spectrum comparison shows YOPO preserves leading eigenvalues better than random pruning.

**Residual Influence Alignment.** Spectral concentration alone does not imply functional relevance. We therefore analyze whether residual magnitude aligns with parameter influence at initialization. Let $\theta$ denote the parameter vector and $f_\theta(x)$ the network output. The diagonal entries of the neural tangent kernel (NTK),

$$\kappa_i = \mathbb{E}_x \left\| \frac{\partial f_\theta(x)}{\partial \theta_i} \right\|^2, \tag{15}$$

measure parameter-wise sensitivity.

For each convolutional layer, we compute the correlation between saliency scores $S_{ij}^{(\ell)}$ and the corresponding NTK diagonal entries. Aggregating across layers yields an average Pearson correlation of approximately 0.28 and an average Spearman correlation of approximately 0.42. The stronger monotonic (Spearman) correlation indicates that parameters with larger residual magnitude consistently rank higher in functional influence, even when the relationship is not strictly linear.

This alignment suggests that residual magnitude captures more than structural deviation from low-rank templates; it is statistically associated with initialization-time sensitivity. In other words, parameters that are poorly explained by dominant additive modes tend also to contribute more strongly to output variation.

**Kernel Strength Preservation.** While the previous analysis addresses local parameter influence, we also examine global kernel strength. The NTK trace,

$$\mathrm{Tr}(\Theta) = \sum_i \kappa_i, \tag{16}$$

provides a measure of total sensitivity retained after pruning. At high sparsity ($\sim 95\%$), all pruning methods substantially reduce NTK trace relative to the dense network, as expected. However, among pruned subnetworks, YOPO consistently preserves slightly higher trace than random pruning and performs comparably to magnitude-based approaches. Although absolute differences in trace are modest, when combined with the observed residual–diagonal alignment, this indicates selective retention of influential parameters rather than indiscriminate scaling of sensitivity.

**Extreme Sparsity and Effective Graph Structure.** We further analyze structural behavior at extreme sparsity levels exceeding 99%. Let the pruning mask induce a directed acyclic graph over neurons and connections. We define effective nodes as neurons participating in at least one active input–output path, and effective paths as distinct surviving routes from input to output. Under extreme compression, YOPO retains fewer effective nodes and substantially fewer effective paths than path-balancing or connectivity-driven methods, yet achieves higher or comparable test accuracy Table 6 and Figure 2. The logarithm of effective path

counts is significantly lower under YOPO at matched sparsity. This indicates that residual-based pruning does not attempt to maximize combinatorial path multiplicity; instead, it preserves a smaller set of structurally distinctive routes. Accuracy retention despite reduced graph complexity suggests that performance depends more on preserving influential and non-template-aligned connections than on maintaining a large number of redundant paths.

Additional empirical evidence comparing layer-wise effective node distributions under ERK, Uniform, and residual-based pruning is provided in Appendix E, illustrating how balanced node retention emerges implicitly under YOPO without explicit node–path constraints.

**Synthesis.** Taken together, these analyses reveal a coherent structural pattern. First, YOPO induces stronger spectral concentration, increasing post-pruning compressibility. Second, residual magnitude aligns monotonically with NTK-based parameter influence, linking structural distinctiveness to functional sensitivity. Third, global NTK trace is preserved at levels comparable to leading data-free alternatives. Finally, under extreme sparsity, YOPO maintains accuracy while substantially reducing effective graph complexity.

These observations support the interpretation that residual-based pruning removes template-aligned redundancy while selectively retaining parameters that are both structurally distinctive and functionally influential at initialization.

## 5 Transferability and Once-Only Saliency

The defining feature of residual-based pruning is that saliency is computed once at initialization and is independent of both data and sparsity budget. This decouples mask construction from downstream training tasks and enables systematic reuse across datasets and compression levels. We formalize this property and introduce quantitative indices for transfer.

**Once-only family of masks.** Let $S(\theta_0) = \{S^{(\ell)}\}_{\ell \in \mathcal{L}}$ be YOPO's layerwise saliency computed *once* at initialization (Sec. 3). For any sparsity $p \in [0, 1]$, define the global mask family

$$\mathcal{M}(p) = \left\{ M^{(\ell)}(\tau_\ell(\alpha(p))) \,\Big|\, \ell \in \mathcal{L} \right\}, \quad \text{where} \quad \tau_\ell(\alpha) = \text{median}(S^{(\ell)}) + \alpha \,\text{MAD}(S^{(\ell)}). \tag{17}$$

By monotonicity, there exists $\alpha(p)$ with achieved sparsity $\hat{s}(\alpha(p)) \approx p$; the *ordering* of entries within each $S^{(\ell)}$ is fixed across all $p$. The local variant $\mathcal{M}_{\text{L}}(p)$ uses per-layer $\alpha_\ell(p)$ to meet per-layer quotas. In both regimes, no saliency recomputation is needed when $p$ changes: $m_p = \text{TopK}(S, k(p))$ or equivalently $m_p = \mathbf{1}[S > \tau(p)]$.

**Dataset transfer.** Because $S(\theta_0)$ depends only on $|\theta_0|$ and a low-rank additive template, the same family $\{\mathcal{M}(p)\}_{p \in [0,1]}$ is applicable to any $D$ drawn after initialization. Let training from scratch on $(\theta_0 \odot \mathcal{M}(p), D)$ yield validation risk $\mathcal{R}(p; D)$ and dense risk $\mathcal{R}_{\text{dense}}(D)$. We evaluate transfer via two indices.

**Definition 1** (Dataset-Transfer Index (DTI))**.** *Fix a source dataset $D_{\text{src}}$ and let $\mathcal{M}_{\text{src}}(p)$ denote the mask family constructed from $S(\theta_0)$ (and, if desired, a concrete mask at a particular $p$). For any target $D_{\text{tgt}}$ and budget $p$,*

$$\text{DTI}(D_{\text{src}} \to D_{\text{tgt}}, p) = \mathcal{R}(p; D_{\text{tgt}} \mid \mathcal{M}_{\text{src}}(p)) - \mathcal{R}(p; D_{\text{tgt}} \mid \mathcal{M}_{\text{tgt}}(p)),$$

*i.e., the excess risk (negative of accuracy gap) when reusing the source mask on the target versus a mask calibrated on the target. Values near 0 indicate successful transfer. We also report the worst-case $\sup_{p \in \mathcal{P}} \text{DTI}(\cdot, p)$ over a budget set $\mathcal{P}$.*

**Definition 2** (Initialization-Independence Index ($\text{I}^3$))**.** *Let $\{\theta_0^{(s)}\}_{s=1}^S$ be $S$ independent random initializations and $\mathcal{M}^{(s)}(p)$ the corresponding masks from the same $S(\cdot)$ construction. For a fixed dataset $D$ and budget $p$,*

$$\text{I}^3(D, p) = \frac{1}{S} \sum_{s=1}^S \left( \mathcal{R}(p; D \mid \mathcal{M}^{(s)}(p)) - \mathcal{R}(p; D \mid \mathcal{M}^{(\bar{s})}(p)) \right),$$

*where $\bar{s}$ denotes applying the mask from seed $s$ to a different initialization. Values near 0 indicate that masks transfer across seeds (sanity for PaI (Su et al., 2020; Ma et al., 2021)).*

**Table 1:** Accuracy (%) and pruning time (seconds) of subnetworks for different pruning methods, models, datasets, and compression ratios. Pruning time is averaged across the four sparsity levels shown. Best results are bolded.

| | RN18 TinyImageNet | | | | | RN20 C10 | | | | | VGG19 C100 | | | | |
|---|---|---|---|---|---|---|---|---|---|---|---|---|---|---|---|
| | Accuracy (%) | | | | Pruning | Accuracy (%) | | | | Pruning | Accuracy (%) | | | | Pruning |
| Sparsity (%) | 68.38 | 90.0 | 96.84 | 99.0 | Time (s) | 68.38 | 90.0 | 96.84 | 99.0 | Time (s) | 68.38 | 90.0 | 96.84 | 99.0 | Time (s) |
| SNIP (Lee et al., 2019) | 56.99 | 53.43 | 48.77 | 36.02 | 5.32 | 87.88 | 84.02 | 76.72 | 62.03 | 1.59 | 71.31 | 70.65 | 67.82 | 61.77 | 4.94 |
| Iter-SNIP (De Jorge et al., 2020) | 56.73 | 53.60 | 48.55 | 36.42 | 232.23 | 88.17 | 84.22 | 77.05 | 64.95 | 55.61 | 72.84 | 70.86 | 67.72 | 63.13 | 116.46 |
| SynFlow (Tanaka et al., 2020) | 56.71 | 54.68 | 49.03 | 39.79 | 97.03 | 88.64 | 84.94 | 78.22 | 66.05 | 55.31 | 71.63 | 69.18 | 66.98 | 62.11 | 100.86 |
| PHEW (Patil & Dovrolis, 2021) | 58.09 | 55.93 | 50.81 | 40.54 | 1912.31 | 90.38 | 87.41 | 81.05 | 70.44 | 25.94 | 73.18 | 70.70 | 68.18 | 64.43 | 2412.82 |
| NPB (Pham et al., 2023) | 58.39 | 56.82 | 51.37 | 41.05 | 382.09 | 90.69 | 87.61 | 80.55 | 70.70 | 21.66 | 74.05 | 71.76 | 68.87 | 64.82 | 426.55 |
| DPaI (Xiang et al., 2025) | 58.70 | 57.30 | **51.57** | 44.30 | 88.77 | 90.93 | 87.79 | **81.50** | **71.50** | 77.12 | 75.00 | 73.23 | 70.13 | 64.80 | 71.80 |
| **YOPO (ours)** | **58.90** | **57.48** | 51.47 | **45.05** | **3.29** | **91.10** | **87.81** | **81.50** | 71.00 | **0.96** | **75.40** | 72.31 | **71.24** | **65.76** | **3.16** |

**Table 2:** Accuracy vs. Sparsity Comparison: YOPO vs. Expanders-based Data-Free Pruning at Initialization (PaIs). Top-1 accuracy (%) at the indicated budgets.

| | | VGG16 | VGG16 | RN18 | RN34 | RN50 | RN101 |
|---|---|---|---|---|---|---|---|
| Method | Topology/Notes | C10 | C100 | | Tiny-ImageNet | | |
| Sparsity (%) | | 96.38 | 99.22 | 96.38 | 99.22 | 96.55 | 97.77 | 85.76 | 91.48 |
| Unpruned Baseline | | 94.24 | 94.24 | 74.16 | 74.16 | 53.88 | 57.08 | 60.13 | 61.29 |
| Random (Liu et al., 2022) | within-layer random | 91.30 | 85.81 | 66.58 | 56.98 | 44.02 | 47.34 | 46.77 | 49.05 |
| X-Net (Prabhu et al., 2018) | d-left-regular | 91.38 | 86.06 | 66.81 | 56.69 | 42.69 | 46.97 | 45.36 | 49.47 |
| RReg (Stewart et al., 2023) | d-regular | 91.50 | 87.02 | 67.72 | 59.61 | 44.30 | 46.30 | 48.27 | 51.21 |
| **YOPO (ours)** | low rank residual | **93.04** | **89.71** | **69.10** | **63.23** | **50.40** | **52.02** | **53.51** | **55.10** |

**Why transfer can hold.** Three structural aspects promote transfer under YOPO: (i) *Architecture-driven statistics*: at random initialization, early convolutional blocks and specific receptive-field patterns exhibit systematically larger Frobenius residuals than later blocks; global calibration preserves these high-residual regions regardless of dataset (Sec. 3). (ii) *Positivity and additivity*: nonnegativity forbids cancellation, so residuals reflect entrywise mismatch from a small additive template that is intrinsic to the parameterization, not to data. (iii) *Nestedness across budgets*: the survivor sets for $p_1 < p_2$ are nested (Proposition 2); moving across budgets is a deterministic re-thresholding, not a rescoring, which stabilizes training. These stand in contrast to data-dependent PaI, where gradients/Hessians encode $D$-specific statistics by construction (Lee et al., 2019; Wang et al., 2020).

# 6 Experiments & Results

In this section, we evaluate sparse subnetworks produced by YOPO by re-training them from scratch on CIFAR-10, CIFAR-100, and Tiny-ImageNet, aligning the experimental setting with prior state-of-the-art pruning-at-initialization (PaI) works. [6] We additionally evaluate on ImageNet-1K (Deng et al., 2009) to verify scalability to large-scale settings. For data-free PaI comparisons, we follow the training configuration used in (Xiang et al., 2025). For data-dependent PaI baselines, we follow the setup of ProsPr (Alizadeh et al., 2022). Unless otherwise specified, all subnetworks are trained using the same optimization schedule as their dense counterparts.

## 6.1 Comparison with Previous SOTA

Table 1 compares YOPO with representative single-shot and iterative PaI methods, including SNIP, Iter-SNIP, SynFlow, PHEW, NPB, and DPaI, across multiple architectures and sparsity levels.

**Accuracy across sparsity regimes.** At moderate sparsity (68.38% and 90%), YOPO achieves the highest accuracy in all three settings, including 58.90% and 57.48% on ResNet-18 (Tiny-ImageNet), 91.10% and 87.81% on ResNet-20 (CIFAR-10), and 75.40% and 72.31% on VGG-19 (CIFAR-100). At extreme sparsity (99%), YOPO attains the highest accuracy on ResNet-18 (45.05%) and VGG-19 (65.76%), and remains

---

[6]Implementation details, training protocols, and reproducibility resources are provided in Appendix K.

**Table 3:** Dataset Transfer Index (DTI) at 95% sparsity. Lower |DTI| indicates better transferability of pruning masks across datasets.

| Method | Model | #Params | C10 | C100 | TinyIN | Mean \|DTI\| | Worst \|DTI\| |
|---|---|---|---|---|---|---|---|
| SNIP (Lee et al., 2019) | VGG16 | 14M | -0.96 | -1.21 | -9.41 | 3.86 | 9.41 |
| GraSP (Wang et al., 2020) | VGG16 | 14M | -0.67 | -0.77 | -8.11 | 3.18 | 8.11 |
| Random | VGG16 | 14M | -0.55 | -0.71 | -3.41 | 1.56 | 3.41 |
| Synflow (Tanaka et al., 2020) | VGG16 | 14M | -0.34 | -0.41 | -1.91 | 0.89 | 1.91 |
| **YOPO (ours)** | VGG16 | 14M | **0.07** | **-0.40** | **-0.21** | **0.23** | **0.40** |
| SNIP (Lee et al., 2019) | ResNet18 | 11M | 0.09 | -0.15 | -5.22 | 1.82 | 5.22 |
| GraSP (Wang et al., 2020) | ResNet18 | 11M | -0.19 | -0.86 | -4.72 | 1.92 | 4.72 |
| Random | ResNet18 | 11M | -0.14 | -0.46 | -2.52 | 1.04 | 2.52 |
| Synflow (Tanaka et al., 2020) | ResNet18 | 11M | -0.04 | -0.26 | -1.22 | 0.51 | 1.22 |
| **YOPO (ours)** | ResNet18 | 11M | **0.11** | **0.02** | **-0.11** | **0.08** | **0.11** |

competitive on ResNet-20 (71.00%). Importantly, these results are obtained without using data or gradients at initialization, distinguishing YOPO from data-dependent methods such as SNIP and ProsPr. A direct comparison with strong data-dependent PaI methods, including ProsPr, is provided in Appendix O, where YOPO remains competitive at moderate sparsity and maintains a clear advantage under extreme compression.

**Pruning time and efficiency.** YOPO exhibits substantially lower pruning overhead than iterative or optimization-based approaches. Across all models, pruning requires only 0.96–3.29 seconds, compared to tens or hundreds of seconds for SynFlow, NPB, and DPaI, and over 1900 seconds for PHEW in some settings. This reduction follows directly from the once-only saliency construction and the nested mask property (Proposition 2), which eliminates the need for iterative rescoring. Detailed FLOPs comparisons across sparsity levels are reported in Appendix F.

## 6.2 Comparison with Connectivity-Based Zero-Shot Methods

We further compare YOPO against expander-style, topology-driven data-free PaI methods (Stewart et al., 2023). Table 2 reports results at extreme sparsities. Expander masks encode strong graph-theoretic priors that preserve signal propagation under high compression. However, degree constraints operate uniformly within a layer and do not provide entrywise selection. In contrast, YOPO defines an elementwise saliency derived from low-rank residual magnitude, enabling fine-grained pruning while retaining global structural coherence. At matched ultra-high sparsities, YOPO consistently outperforms Random, X-Net, and RReg on VGG-16 (CIFAR-10/100), ResNet-34 (Tiny-ImageNet), and ImageNet ResNet-50. These gains are achieved without data or gradient information, demonstrating that residual-based structural selection can outperform purely topological wiring schemes.

## 6.3 Large-Scale Evaluation on ImageNet

**Table 4:** Comparison of Avg and Best Acc(%) between Synflow and DPaI Methods on ImageNet-1K

| | Avg Acc(%) | Best Acc(%) |
|---|---|---|
| Synflow (Tanaka et al., 2020) | $71.4 \pm 0.29$ | 71.8 |
| DPaI (Xiang et al., 2025) | $72.2 \pm 0.25$ | 72.5 |
| YOPO (ours) | $\textbf{73.2} \pm \textbf{0.15}$ | **73.4** |

To assess scalability, we evaluate YOPO on ImageNet-1K. Table 4 compares average and best Top-1 accuracy with SynFlow and DPaI. YOPO achieves an average Top-1 accuracy of 73.2% and a best accuracy of 73.4%, outperforming both SynFlow and DPaI under identical training settings. These results confirm that the

residual-based saliency remains effective in large-scale regimes. Additional results on modern architectures such as ConvNeXt backbones under moderate sparsity are provided in Appendix G.

## 6.4 Extreme Sparsity and Structural Efficiency

We analyze structural behavior under sparsity exceeding 99%, measuring effective nodes and logarithmic effective path counts. Despite preserving fewer effective nodes and substantially fewer paths than DPaI and NPB, YOPO maintains higher or comparable accuracy at matched sparsity. This finding indicates that combinatorial path multiplicity alone does not guarantee performance. Instead, selectively retaining structurally distinctive parameters-those poorly explained by dominant additive templates-appears more important than maximizing connectivity.

**Layer-wise effective nodes and paths.**   We compare YOPO against ERK and Uniform layer-wise sparsity under extreme compression. Uniform allocation collapses at 99.90% sparsity, whereas both ERK and YOPO remain trainable. However, ERK preserves substantially more effective paths (e.g., log paths 44.98 vs. 28.57 at 99.68%), yet achieves lower accuracy (30.88% vs. 35.95%). This indicates that combinatorial path multiplicity alone does not determine performance. YOPO maintains a harmoniously distributed set of effective nodes across stages without explicitly enforcing node–path balance, suggesting that residual-based structural selectivity implicitly induces a well-conditioned subnetwork. For a full layer-wise analysis of effective node distributions and path counts across ERK, Uniform, and YOPO, see Appendix E.

## 6.5 Budget Reuse and Dataset Transfer

YOPO decouples parameter *ordering* from the sparsity *budget* (Section 5). After computing the residual-based saliency $S^{(\ell)}$ at initialization, all sparsity levels are obtained through monotone re-thresholding without rescoring (Proposition 2). This produces a nested family of masks and enables stable reuse across $\{50, 70, 80, 90, 95, 98\}\%$ sparsities (Appendix Table 15).

We further evaluate cross-dataset mask transfer. Transferability is quantified using the Dataset-Transfer Index (DTI; Definition 1), defined as the excess risk incurred when a mask constructed on a source dataset is reused on a target dataset at identical sparsity. Table 3 reports results at 95% sparsity for VGG16 and ResNet18. YOPO achieves consistently near-zero DTI across architectures and datasets. For VGG16 at 95% sparsity, YOPO obtains mean $|DTI| = 0.23$ and worst-case $|DTI| = 0.40$, substantially lower than SynFlow (0.89 / 1.91), Random (1.56 / 3.41), GraSP (3.18 / 8.11), and SNIP (3.86 / 9.41). On ResNet18, YOPO attains mean $|DTI| = 0.08$ and worst-case $|DTI| = 0.11$, again outperforming competing methods. The degradation pattern highlights the role of data dependence. Methods such as SNIP and GraSP rely on gradients or Hessians evaluated on a specific dataset, causing their masks to encode dataset-specific statistics. When transferred from CIFAR-10 to TinyImageNet, these methods incur substantial accuracy drops (up to 9.41 points). SynFlow, while data-free, remains iterative and still exhibits larger transfer gaps than YOPO.

In contrast, YOPO saliency depends only on weight geometry and the additive low-rank structure present at initialization, making the resulting masks independent of the dataset by construction. The results in Table 3 therefore indicate that once-only residual-based scoring yields masks that remain stable across both sparsity budgets and datasets, even at extreme sparsity.

## 6.6 Discussion: NTK Preservation and Structural Selectivity

Although absolute NTK trace differences among pruned models are modest, trace alone is a coarse scalar summary of the full kernel spectrum. Optimization dynamics are governed primarily by dominant eigen-directions rather than total trace magnitude. Our analysis in Section 4 shows that YOPO exhibits stronger alignment between residual magnitude and NTK-based parameter influence (Spearman $\approx 0.42$) Table O, indicating selective preservation of functionally important directions. Therefore, the advantage of YOPO does not stem from marginal trace improvements, but from structurally informed retention of influential NTK components while removing template-aligned redundancy. Further analysis of retained residual distributions is presented in Appendix H.

**Ablations and stability.** We evaluate the sensitivity of YOPO to the low-rank factorization parameter, thresholding scheme (MAD vs. STD), minimal keep constraints, per-channel versus full-layer NMF, and initialization choice. Across architectures and datasets, performance remains stable over a broad range of ranks and sparsity budgets, with global thresholding consistently outperforming layer-wise allocation at fixed settings. Residual-based saliency is robust to initialization and exhibits strong cross-seed consistency, and inverse-residual scoring confirms that high-residual weights encode structurally meaningful information. Detailed analyses, figures, and tables are provided in Appendix I.

## 7 Conclusion

We introduced YOPO, a zero-shot, data and gradient-free pruning-at-initialization method based on nonnegative low-rank residual saliency. By computing a once-only ordering of parameters at random initialization, YOPO decouples parameter ranking from sparsity budget and dataset, enabling monotone re-thresholding without rescoring.

Our analysis shows that residual-based pruning aligns structural redundancy reduction with effective connectivity preservation and training dynamics. Spectral evidence supports a low-rank template structure in randomly initialized weights, while empirical results demonstrate that YOPO maintains competitive or superior accuracy across sparsity regimes, including extreme compression levels above 99%. At the same time, YOPO significantly reduces pruning time relative to optimization-driven approaches and remains effective on modern architectures such as ConvNeXt.

Together, these findings suggest that meaningful sparse subnetworks can be identified from intrinsic structural properties of random initialization alone. More broadly, our results highlight the importance of understanding initialization structure in determining sparse trainability, and point toward a unifying perspective linking low-rank redundancy, connectivity balance, and optimization dynamics in pruning at initialization. Limitations and broader implications of this work are discussed in Appendix D.

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

## A  Collapse Avoidance: Positivity & Robust Thresholds

We formalize conditions under which YOPO avoids neuron/row collapse in a *single shot*. Recall $S^{(\ell)} = \left| \tilde{W}^{(\ell)} - V^{(\ell)} H^{(\ell)} \right| \in \mathbb{R}_{\geq 0}^{o_\ell \times d_\ell}$, $\tau_\ell(\alpha)$ is a robust layer-wise threshold (MAD or STD), and $M^{(\ell)}(\alpha) = \mathbf{1}\left[ S^{(\ell)} > \tau_\ell(\alpha) \right]$ is the binary mask. Let $S_{i\cdot}^{(\ell)}$ denote the $i$-th row of $S^{(\ell)}$ and $\| \cdot \|_\infty$ the max norm.

**Proposition 3** (Row survival under positive reconstruction). *Fix a layer $\ell$. Suppose $\tilde{W}^{(\ell)} \geq 0$ and $V^{(\ell)} H^{(\ell)} \geq 0$. If a row $i$ satisfies $\tilde{W}_{i\cdot}^{(\ell)} \neq \left( V^{(\ell)} H^{(\ell)} \right)_{i\cdot}$, then $\| S_{i\cdot}^{(\ell)} \|_\infty > 0$. Consequently, for any threshold $\tau_\ell$ with $\tau_\ell < \| S_{i\cdot}^{(\ell)} \|_\infty$ we have $\sum_{j=1}^{d_\ell} M_{ij}^{(\ell)} > 0$, i.e., row $i$ retains at least one nonzero parameter.*

**Discussion.**   Positivity eliminates sign cancellation: any coordinate-wise mismatch between $\tilde{W}_{i\cdot}^{(\ell)}$ and its nonnegative reconstruction must manifest as a strictly positive residual on that coordinate, guaranteeing a nonzero max-residual. Thus, *whenever* the chosen layer threshold lies below this max-residual, the row

cannot vanish. In practice, (i) exact equality $\tilde{W}_{i\cdot}^{(\ell)} = \left(V^{(\ell)}H^{(\ell)}\right)_{i\cdot}$ is a measure-zero event under continuous random initializations and finite-rank NMF; (ii) we adopt strict inequality in the mask, $M_{ij}^{(\ell)} = \mathbf{1}[S_{ij}^{(\ell)} > \tau_\ell]$, to break ties; and (iii) we enforce a minimal per-row keep:

$$\sum_{j=1}^{d_\ell} M_{ij}^{(\ell)} \geq m_{\min} \qquad (\text{typically } m_{\min} \in \{1, 2\}), \tag{18}$$

which acts as a deterministic safeguard even in degenerate numerical cases. We provide the proofs in Appendix A.1.

**Corollary 1** (No layer collapse). *If at least one row $i$ satisfies $\tilde{W}_{i\cdot}^{(\ell)} \neq \left(V^{(\ell)}H^{(\ell)}\right)_{i\cdot}$ and $\tau_\ell < \max_i \|S_{i\cdot}^{(\ell)}\|_\infty$, then $\sum_{i,j} M_{ij}^{(\ell)} > 0$, i.e., the layer retains parameters. Furthermore, with the constraint equation 18, every row retains at least $m_{\min}$ entries.*

**Practical instantiation.** Because $\hat{s}(\alpha)$ is monotone in $\alpha$, bisection selects $\alpha$ to meet a feasible global budget; empirical ranges of $\alpha$ that achieve $s \in [0.1, 0.99]$ invariably satisfy $\tau_\ell < \max_i \|S_{i\cdot}^{(\ell)}\|_\infty$ unless $S^{(\ell)} \equiv 0$ (an event tantamount to perfect NMF reconstruction, Proposition1). In rare near-degenerate cases, the explicit $m_{\min}$ rule guarantees row survival.

## A.1  Proofs for Section A

We collect elementary lemmas used in the proof. All vectors/matrices are real and nonnegative where stated. For a vector $x$, $\|x\|_\infty = \max_j |x_j|$; for a matrix $A$, $A_i$ denotes the $i$-th row.

**Lemma 3** (Positive mismatch implies positive max-residual). *Let $x, \hat{x} \in \mathbb{R}_{\geq 0}^d$ and define $s = |x - \hat{x}|$ elementwise. If $x \neq \hat{x}$, then $\|s\|_\infty > 0$.*

*Proof.* If $x \neq \hat{x}$, there exists $j^\star$ with $x_{j^\star} \neq \hat{x}_{j^\star}$. Then $|x_{j^\star} - \hat{x}_{j^\star}| > 0$, hence $\|s\|_\infty \geq |x_{j^\star} - \hat{x}_{j^\star}| > 0$. $\qquad\square$

**Lemma 4** (Row-wise residual positivity for NMF reconstructions). *Fix a layer $\ell$ and let $\tilde{W}^{(\ell)} \in \mathbb{R}_{\geq 0}^{o_\ell \times d_\ell}$, $V^{(\ell)}H^{(\ell)} \in \mathbb{R}_{\geq 0}^{o_\ell \times d_\ell}$. If $\tilde{W}_{i\cdot}^{(\ell)} \neq \left(V^{(\ell)}H^{(\ell)}\right)_{i\cdot}$, then $\left\| \, |\tilde{W}_{i\cdot}^{(\ell)} - (V^{(\ell)}H^{(\ell)})_{i\cdot}| \, \right\|_\infty > 0$.*

*Proof.* Apply Lemma 3 with $x = \tilde{W}_{i\cdot}^{(\ell)}$ and $\hat{x} = (V^{(\ell)}H^{(\ell)})_{i\cdot}$. $\qquad\square$

**Lemma 5** (Strict-threshold survival). *Let $s \in \mathbb{R}_{\geq 0}^d$ and $\tau \geq 0$. If $\tau < \|s\|_\infty$ and the mask is defined by $m_j = \mathbf{1}[s_j > \tau]$, then $\sum_{j=1}^d m_j \geq 1$.*

*Proof.* Let $j^\star \in \arg\max_j s_j$; then $s_{j^\star} = \|s\|_\infty > \tau$, so $m_{j^\star} = 1$. $\qquad\square$

*Proof of Proposition 3.* Fix a row $i$. If $\tilde{W}_{i\cdot}^{(\ell)} \neq (V^{(\ell)}H^{(\ell)})_{i\cdot}$, then by Lemma 4, $\|S_{i\cdot}^{(\ell)}\|_\infty > 0$. For any $\tau_\ell < \|S_{i\cdot}^{(\ell)}\|_\infty$, Lemma 5 with $s = S_{i\cdot}^{(\ell)}$ ensures $\sum_j M_{ij}^{(\ell)} = \sum_j \mathbf{1}[S_{ij}^{(\ell)} > \tau_\ell] \geq 1$. $\qquad\square$

**On degeneracy and probability-one statements.** If $\tilde{W}^{(\ell)} = (V^{(\ell)}H^{(\ell)})$ exactly, then $S^{(\ell)} \equiv 0$ and any $\tau_\ell \geq 0$ yields $M^{(\ell)} \equiv 0$. Under continuous random initializations and finite rank $r_\ell < \min(o_\ell, d_\ell)$, the event of exact equality has Lebesgue measure zero (heuristically, it requires solving a system of polynomial equalities $o_\ell d_\ell$ constraints with only $r_\ell(o_\ell + d_\ell)$ degrees of freedom). In practice, finite-precision multiplicative updates do not produce exact equality, and our implementation additionally enforces equation 18, guaranteeing per-row survival regardless of numerical coincidences.

**Choice of threshold and ties.** We adopt a strict indicator $\mathbf{1}[S_{ij}^{(\ell)} > \tau_\ell]$ to avoid ambiguity when an entry equals the threshold. For robust thresholds (median/MAD or mean/STD), $\tau_\ell < \max_{i,j} S_{ij}^{(\ell)}$ whenever $S^{(\ell)}$ is not constant, ensuring that at least some entries (and often each nontrivial row) survive without resorting to the $m_{\min}$ safeguard.

# B  Methodology

We highlight the YOPO pruning process in Algorithm 1

---

**Algorithm 1** YOPO: You Only Prune Once (Pseudocode)

---

**Require:** Random init $\{W^{(\ell)}\}_{\ell \in \mathcal{L}}$; target sparsity $s$ (global) or budgets $\{s_\ell\}$ (local); ranks $\{r_\ell\}$; min-keep $(m_{\min}, c_{\min})$

---

 1: **for** $\ell \in \mathcal{L}$ **do** ▷ For each layer compute the saliency
 2:     $\tilde{W}^{(\ell)} \leftarrow \left| \text{flat}(W^{(\ell)}) \right|$ ▷ optional: layerwise normalization
 3:     Solve NMF equation 6 for $V^{(\ell)}, H^{(\ell)}$
 4:     $S^{(\ell)} \leftarrow \left| \tilde{W}^{(\ell)} - V^{(\ell)} H^{(\ell)} \right|$
 5: **end for**
 6: **if** global (YOPO-G) **then** ▷ Select the sparsity type
 7:     Find $\alpha$ by MORT (Alg. 2), so that $s(\{M^{(\ell)}(\alpha)\}) \approx s$
 8: **else** ▷ local (YOPO-L)
 9:     Choose $\alpha_\ell$ by MORT (Alg. 2), so that per-layer budgets $s_\ell$ hold
10: **end if**
11: **for** $\ell \in \mathcal{L}$ **do** ▷ Apply the once-only mask
12:     $\tau_\ell \leftarrow \text{median}(S^{(\ell)}) + \alpha \, \text{MAD}(S^{(\ell)})$ ▷ or $\mu + \alpha\sigma$
13:     $M^{(\ell)} \leftarrow \mathbf{1}\left[ S^{(\ell)} > \tau_\ell \right]$; enforce row/col keep using equation 10
14:     Apply $M^{(\ell)}$; freeze gradients: $\nabla W^{(\ell)} \leftarrow \nabla W^{(\ell)} \odot \text{unflat}(M^{(\ell)})$
15: **end for**
16: Train the masked model from scratch under a same recipe

---

# C  Related Work

Pruning at initialization (PaI) emerged from the Lottery Ticket Hypothesis (LTH), which demonstrated that dense randomly initialized networks contain sparse subnetworks capable of matching dense performance when trained in isolation (Frankle & Carbin, 2019). Subsequent work strengthened this perspective by studying rewinding strategies (Frankle et al., 2020), dynamic sparse training (Evci et al., 2020; Jayakumar et al., 2020), and structural characterizations of strong lottery tickets (Ramanujan et al., 2020; Natale et al., 2024). Sanity-check analyses further clarified the role of randomness and initialization in sparse trainability (Su et al., 2020; Ma et al., 2021; Liu et al., 2022).

**Gradient- and data-dependent PaI.** Early practical PaI criteria relied on gradient-based sensitivity measures. SNIP prunes connections based on first-order loss sensitivity at initialization (Lee et al., 2019), while GraSP preserves gradient flow through second-order approximations (Wang et al., 2020). ProsPr uses meta-gradients to identify trainable weights before training (Alizadeh et al., 2022). Although effective, these approaches require data or backpropagation at initialization and are typically coupled to a specific sparsity budget.

**Data-agnostic and topology-driven PaI.** To remove dependence on training data, SynFlow introduced iterative flow conservation to avoid layer collapse (Tanaka et al., 2020). PHEW and related methods construct sparse networks through structural heuristics without training data (Patil & Dovrolis, 2021). More recent approaches explicitly optimize connectivity properties. Node–path balance principles argue that trainability depends on maintaining equilibrium between effective nodes and effective paths (Pham et al., 2023; Xiang et al., 2025). Expander-inspired constructions impose graph-theoretic connectivity constraints to preserve information flow (Stewart et al., 2023; Prabhu et al., 2018; Hoory et al., 2006). Connectivity-based analyses show that effective sparsity, rather than raw parameter count, governs performance under extreme compression (Vysogorets & Kempe, 2023). Restricted random pruning further explores structural allocation strategies to extend feasible sparsity ranges (Otsuka et al., 2024).

**Structural perspectives on sparse trainability.** Several works suggest that sparse trainability is governed by intrinsic structural properties of random networks. Rare Gems identifies promising subnetworks directly at initialization (Sreenivasan et al., 2022), while graph transformer studies uncover strong lottery tickets linked to architectural structure (Ito et al., 2025). Theoretical analyses also identify information-theoretic limits to pruning at initialization (Kumar et al., 2024). Broader surveys synthesize the landscape of pruning methodologies and highlight the trade-offs between structural constraints, computational cost, and generalization (Wang et al., 2022; Cheng et al., 2024).

**Low-rank structure and initialization dynamics.** Spectral analyses of deep networks show that random weight matrices exhibit structured singular value behavior that influences optimization (Pennington et al., 2017). Neural tangent kernel (NTK) theory formalizes how initialization governs training dynamics in overparameterized regimes (Jacot et al., 2018). Nonnegative matrix factorization (NMF) provides a principled mechanism for decomposing additive components in nonnegative matrices (Lee & Seung, 1999; 2000). These results suggest that dominant low-rank templates may already be present at initialization, and that deviations from such templates could encode structurally distinctive or influential parameters.

**Positioning of YOPO.** YOPO differs from gradient-based methods by remaining fully data and gradient-free. Unlike topology-driven approaches that explicitly optimize node–path objectives or impose graph constraints, YOPO derives sparsity from a once-only low-rank residual ordering without solving discrete or differentiable structural programs. This decouples parameter ranking from sparsity budget and dataset, enabling mask reuse through monotone re-thresholding. Conceptually, YOPO bridges structural redundancy detection and dynamical influence alignment by leveraging low-rank template deviation as a saliency signal.

# D    Limitations and Implications

While YOPO provides a once-only, data-free pruning criterion with strong empirical performance, several limitations merit careful consideration.

First, YOPO performs unstructured pruning. Although theoretical FLOPs decrease proportionally with sparsity, practical acceleration on modern hardware is limited without structured sparsity support. While we observe emergent kernel-level structure, this behavior is not enforced and may vary across architectures. Translating residual-based saliency into hardware-aligned structured pruning remains an open challenge. Second, the method relies on nonnegative low-rank factorization of absolute weights. While we provide structural and dynamical evidence supporting this design, the precise theoretical relationship between residual magnitude and long-term optimization dynamics remains incompletely characterized. Recent information-theoretic analyses suggest inherent limits to pruning at initialization, and a deeper theoretical treatment of when low-rank residual structure suffices would be valuable. Third, our experiments focus primarily on convolutional architectures and vision benchmarks. Although preliminary results on modern architectures such as ConvNeXt indicate generality, extending the analysis to transformers, multimodal models, and large-scale foundation models would further clarify the scope of applicability. Fourth, YOPO does not explicitly optimize global structural properties such as node–path balance. Although it implicitly achieves favorable structural allocation at extreme sparsity, there is no guarantee that this behavior persists under all architectures or depth scales. In particular, behavior under very deep networks or highly anisotropic layer dimensions warrants further analysis. Finally, YOPO assumes standard random initialization schemes. Understanding how alternative initialization distributions or normalization strategies affect the low-rank template structure may provide further insights into the mechanism underlying initialization-time sparsification.

Looking forward, several directions are promising. Extending residual-based saliency toward structured sparsity could improve deployment efficiency. Developing theoretical analyses that relate low-rank template deviation to optimization dynamics would clarify the conditions under which once-only pruning succeeds. Evaluating YOPO at transformer scale and in multimodal or foundation-model settings would further test the generality of the approach. More broadly, understanding how low-rank redundancy at initialization shapes sparse trainability may inform both pruning strategies and initialization design.

# E    Layer-wise Effective Nodes and Paths Under Extreme Sparsity

To further analyze structural behavior at extreme sparsity, we compare YOPO with ERK and Uniform layer-wise sparsity using ResNet18 on Tiny-ImageNet. We report layer-wise effective node counts, logarithmic effective path counts, and test accuracy at 99.00%, 99.68%, and 99.90% sparsity.

**Table 5:** Layer-wise effective nodes and effective paths under extreme sparsity on ResNet18–TinyImageNet.

| Sparsity | Method | Layer-wise effective nodes | log Paths | Acc (%) |
|---|---|---|---|---|
| 99.00% | ERK | 64-64-64-64-128-128-128-128-256-256-256-256-512-512-512-512-512-512-200 | 62.47 | 44.93 |
| | Uniform | 4-37-50-51-60-117-126-55-126-128-255-256-210-256-256-512-503-512-351-200 | 64.94 | 41.01 |
| | YOPO | 64-64-64-64-64-128-128-128-128-128-256-256-256-256-256-512-512-512-512-200 | **46.22** | **45.15** |
| 99.68% | ERK | 64-64-64-64-64-128-128-128-252-256-256-256-256-502-512-511-506-497-200 | 44.98 | 30.88 |
| | Uniform | 6-18-22-16-21-32-66-8-56-67-144-231-63-210-217-468-503-248-482-88-200 | 66.11 | 16.33 |
| | YOPO | 64-64-64-64-64-128-128-128-128-128-256-256-256-256-512-511-511-512-478-200 | **28.57** | **35.95** |
| 99.90% | ERK | 41-35-50-37-59-68-114-123-77-113-140-215-238-143-215-257-385-414-285-296-197 | 49.99 | 16.33 |
| | Uniform | 0-0-0-0-0-0-0-0-0-0-0-0-0-0-0-0-0-0-0-0-0 | − | − |
| | YOPO | 53-55-54-58-59-82-64-65-83-89-161-128-108-150-151-262-256-187-232-335-200 | **29.82** | **16.75** |

**Table 6:** Extreme sparsity analysis comparing structural properties and test accuracy. We report the number of effective nodes, log effective paths, and final test accuracy (mean ± std over 3 runs).

| Method | Sparsity | Effective Nodes | log Effective Paths | Test Acc. (%) |
|---|---|---|---|---|
| YOPO | 99.68% | $4072 \pm 10$ | $28.57 \pm 0.31$ | **35.97 ± 0.34** |
| | 99.90% | $2866 \pm 42$ | $29.82 \pm 0.74$ | **16.75 ± 0.47** |
| DPaI (Xiang et al., 2025) | 99.68% | $4974 \pm 7$ | $45.12 \pm 0.24$ | $30.73 \pm 0.27$ |
| | 99.90% | $3496 \pm 203$ | $49.96 \pm 0.93$ | $15.69 \pm 0.37$ |
| NPB (Pham et al., 2023) | 99.68% | $500 \pm 23$ | $76.15 \pm 0.90$ | $24.33 \pm 0.19$ |
| | 99.90% | $626 \pm 32$ | $64.06 \pm 1.58$ | $11.73 \pm 0.62$ |

Table 5 summarizes the results. Uniform allocation collapses at 99.90% sparsity, with all layers effectively zeroed. Even at 99.68%, several bottleneck layers contain very few effective nodes, leading to degraded accuracy. ERK maintains trainability at extreme sparsity and preserves substantially more effective paths. However, increased combinatorial path counts do not necessarily translate to improved accuracy. YOPO remains trainable at all sparsity levels, maintains harmoniously distributed effective nodes across stages, and achieves higher or comparable accuracy despite preserving fewer effective paths.

These results reinforce the structural selectivity principle: performance depends more on retaining structurally distinctive connections than on maximizing path multiplicity.

# F    Results for FLOPs Analysis

Table 7 reports inference-time FLOPs of subnetworks produced by different pruning-at-initialization (PaI) methods across architectures and sparsity levels. FLOPs are computed as theoretical MAC operations under identical input resolutions and configurations. Since all methods apply unstructured pruning, FLOPs scale approximately with the number of retained parameters.

Across sparsity regimes, YOPO produces subnetworks with FLOPs comparable to topology-driven approaches such as NPB and DPaI, and generally aligned with data-dependent baselines. At moderate sparsity (68.38% and 90%), FLOPs remain nearly indistinguishable across methods. As sparsity increases (96.84% and 99%), YOPO maintains similar or slightly lower FLOPs while preserving stronger accuracy (Table 1).

These results indicate that YOPO does not incur additional computational cost relative to competing PaI criteria. The efficiency gains observed in the main paper stem primarily from reduced pruning time and strong accuracy retention under extreme sparsity, rather than aggressive FLOPs reduction. Overall, YOPO achieves

**Table 7:** FLOPs of subnetworks for different pruning methods, models, datasets, and compression ratios.

| | RN18 TinyImageNet FLOPs ($\times 10^8$) | | | | RN20 CIFAR-10 FLOPs ($\times 10^6$) | | | | VGG19 CIFAR-100 FLOPs ($\times 10^7$) | | | |
|---|---|---|---|---|---|---|---|---|---|---|---|---|
| Sparsity (%) | 68.38 | 90.0 | 96.84 | 99.0 | 68.38 | 90.0 | 96.84 | 99.0 | 68.38 | 90.0 | 96.84 | 99.0 |
| SNIP (Lee et al., 2019) | 11.35 | 5.77 | 3.04 | 1.55 | 17.952 | 8.323 | 3.470 | 1.709 | 17.952 | 7.806 | 3.686 | 1.816 |
| Iter-SNIP (De Jorge et al., 2020) | 10.73 | 7.05 | 3.98 | 1.97 | 18.465 | 9.698 | 4.510 | 2.022 | 18.465 | 9.479 | 4.951 | 2.529 |
| SynFlow (Tanaka et al., 2020) | 14.71 | 8.91 | 4.24 | 1.50 | 22.998 | 11.549 | 4.263 | 1.633 | 22.998 | 12.702 | 6.306 | 2.605 |
| PHEW (Patil & Dovrolis, 2021) | 14.29 | 8.35 | 3.92 | 1.50 | 22.108 | 10.690 | 4.110 | 1.640 | 22.108 | 11.746 | 5.611 | 2.340 |
| NPB (Pham et al., 2023) | 14.37 | 5.21 | 1.74 | 0.59 | 22.035 | 7.642 | 2.645 | 1.122 | 22.035 | 8.773 | 2.874 | 1.046 |
| DPaI (Xiang et al., 2025) | 14.37 | 5.20 | 1.74 | 0.60 | 23.683 | 7.642 | 2.645 | 1.114 | 22.035 | 8.773 | 2.873 | 1.046 |
| YOPO | 14.35 | 5.43 | 1.95 | 0.58 | 14.67 | 5.569 | 1.876 | 0.955 | 18.207 | 8.545 | 2.893 | 1.084 |

computational profiles consistent with topology-aware methods while remaining fully data and gradient-free. In addition to matching the FLOPs profile of competing PaI methods, YOPO induces emergent structure at the kernel level. As shown in Figure 1(a), pruning frequently removes entire 2D convolutional kernels (i.e., complete input–output channel connections), producing structured kernel-level sparsity despite applying an element-wise masking rule. Although YOPO does not explicitly enforce structured sparsity, this emergent behavior suggests a natural pathway toward structured or hardware-aware extensions in future work.

## G   Experiments on Modern Architectures

Table 8 reports fine-tuning results for YOPO-pruned ConvNeXt models on Caltech-256 and Tiny-ImageNet. The total number of prunable parameters is 27,909,600, and we evaluate subnetworks corresponding to moderate (20%) and higher (50%) sparsity levels.

On Caltech-256, pruning 20% of parameters reduces the model to 22,327,680 weights with only a 1.28% absolute accuracy drop relative to the dense baseline (91.62% to 90.34%). At 50% sparsity (13,954,800 parameters), performance decreases to 82.58%, reflecting the expected degradation under substantial compression while remaining competitive for the given parameter budget.

**Table 8:** Fine-tuning YOPO-pruned ConvNeXt on modern architectures. Total prunable parameters: 27,909,600. Accuracy reported in (%).

| ConvNeXt (Caltech-256) | Parameters Reduced To | Accuracy |
|---|---|---|
| Baseline | 27,909,600 | 91.62 |
| YOPO | 22,327,680 | 90.34 |
| YOPO | 13,954,800 | 82.58 |
| **ConvNeXt (Tiny-ImageNet)** | Parameters Reduced To | Accuracy |
| Baseline | 27,909,600 | 86.31 |
| YOPO | 22,327,680 | 84.91 |
| YOPO | 13,954,800 | 78.55 |

A similar trend is observed on Tiny-ImageNet using ImageNet-pretrained weights. At 20% sparsity, accuracy decreases modestly from 86.31% to 84.91%. At 50% sparsity, performance remains at 78.55%, demonstrating stable fine-tuning behavior under structured initialization pruning. These results indicate that YOPO extends beyond classical CNN backbones and remains effective on modern architectures such as ConvNeXt. In particular, moderate sparsity levels incur only minor accuracy degradation after fine-tuning, suggesting that the once-only initialization-time mask preserves transferable and trainable structure even in contemporary large-scale models.

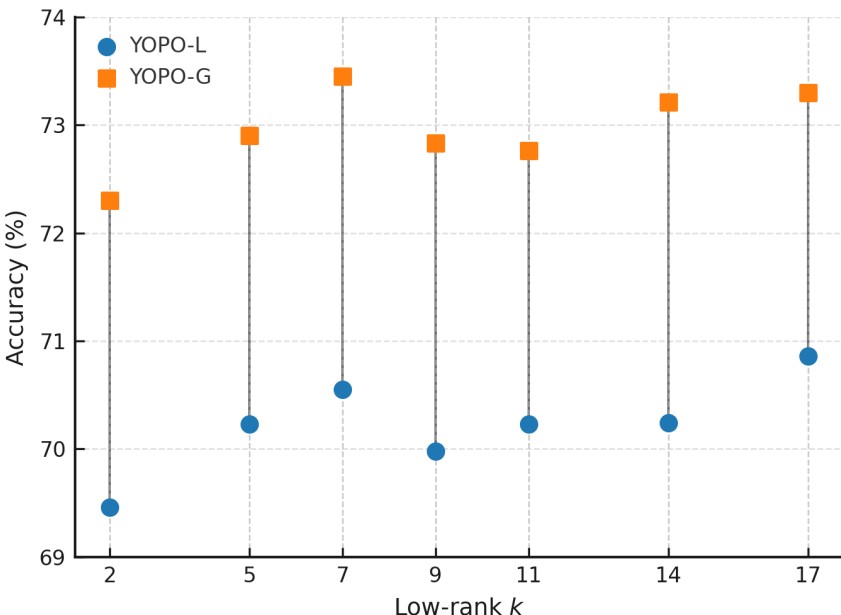

**Figure 4: Impact of low-rank parameter $k$ on test accuracy.** YOPO-G outperforms YOPO-L by $\sim$2–3.5% across ranks under 90% sparsity on CIFAR-100, with both methods remaining stable as $k$ varies.

## H  Analysis of Retained NMF Residuals

We analyze the distribution of NMF residual magnitudes for weights *retained* after pruning by YOPO and compare them to a randomly masked model at matched global sparsity ($\approx 95.04\%$). The goal is to examine whether YOPO selectively preserves structurally distinctive parameters as implied by its residual-based saliency rule.

**Distributional comparison.**  Figure 5 shows kernel density estimates (KDE) of retained residual magnitudes. The YOPO-pruned distribution is shifted toward higher residual values relative to the randomly masked model. Its mode occurs at a substantially larger residual magnitude, and the overall density occupies a higher residual range. In contrast, the random baseline concentrates around smaller residual values and exhibits a narrower spread. NMF residual magnitude exhibits a moderate positive correlation with masked weight magnitude across layers (Pearson: mean 0.51, median 0.52; Spearman: mean 0.48, median 0.46), indicating consistent alignment between residual-based saliency and parameter strength.

**Statistical summaries.**  The summary statistics further quantify this separation. For the retained weights:

| Model | Mean | Median | Std. Dev. |
|---|---|---|---|
| YOPO-pruned | 0.0368 | 0.0338 | 0.0106 |
| Random masked | 0.0108 | 0.0094 | 0.0087 |

Both the mean and median residual magnitudes are substantially larger for YOPO-retained weights. This indicates that YOPO preferentially preserves parameters that are less well-approximated by the low-rank NMF basis.

**Consistency with the pruning mechanism.**  YOPO assigns higher saliency to parameters with larger residuals, i.e., weights that deviate most from the learned nonnegative low-rank template. By pruning low-residual weights, the algorithm removes parameters that are well-explained by the dominant low-rank

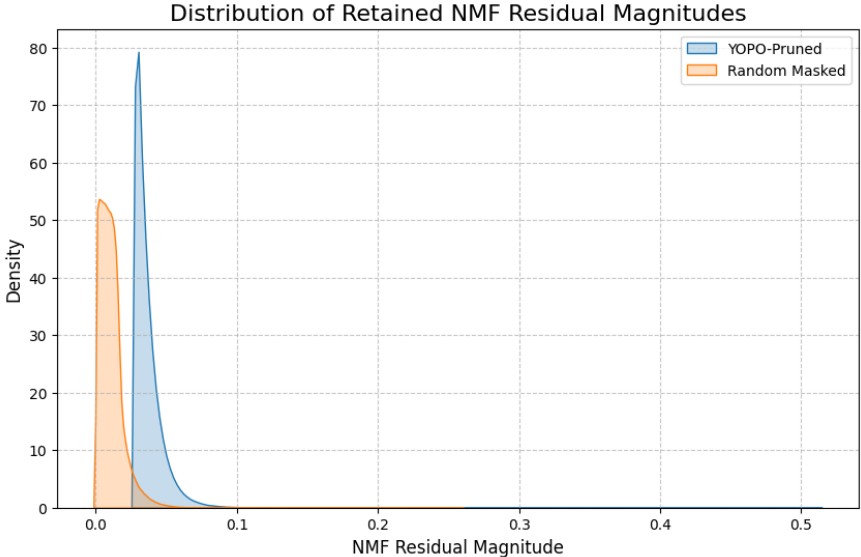

**Figure 5:** Distribution of retained NMF residual magnitudes at $\approx 95.04\%$ sparsity. YOPO retains predominantly high-residual weights, whereas random masking retains many low-residual weights aligned with the low-rank template.

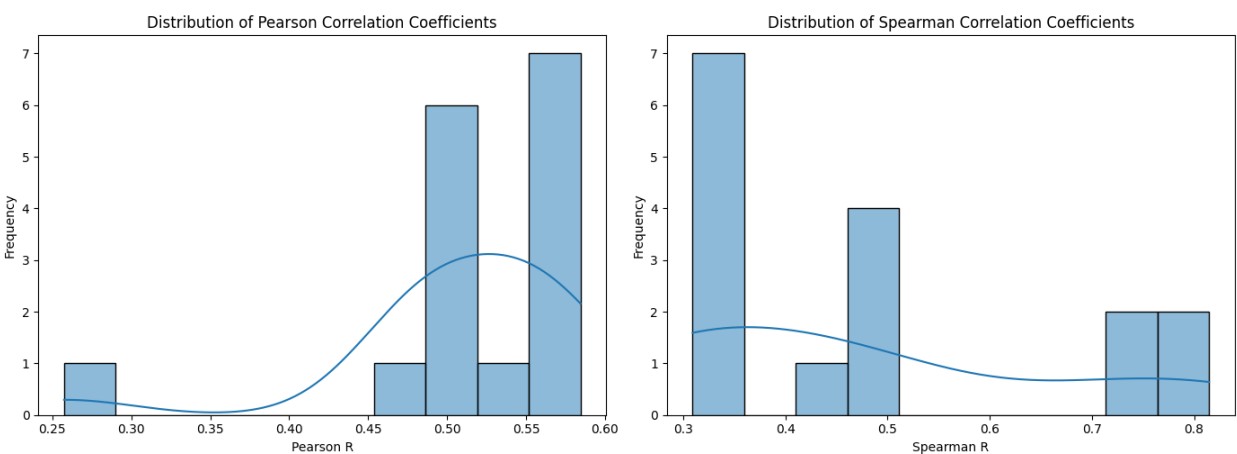

**Figure 6:** Layer-wise distribution of Pearson and Spearman correlations between NMF residual magnitudes and masked weight magnitudes. Both metrics show moderate positive association across layers, with variability reflecting structural heterogeneity within the network.

structure. The retained set therefore naturally exhibits higher residual magnitudes. The empirical separation in Fig. 5 confirms that this mechanism operates as intended and is not merely an artifact of thresholding.

**Relation to the low-rank template hypothesis.** These findings support the low-rank template hypothesis: random convolutional weight tensors admit a dominant additive low-rank structure, and a substantial fraction of parameters are redundant with respect to this template. By removing weights that conform closely to the low-rank approximation and retaining those that deviate from it, YOPO preserves structurally distinctive components beyond the dominant modes. In contrast, random pruning retains a mixture of low- and high-residual weights, including many that are already well-explained by the low-rank structure.

**Visualization of NMF components.**   Figure 7 shows heatmaps of the learned NMF factors $(V, H)$ for a representative VGG-19 layer. The basis matrix $V$ captures how each low-rank component contributes to the reconstruction of individual output channels, while the activation matrix $H$ reflects how these components are expressed across the flattened input features. Together, these visualizations illustrate the structured, component-wise decomposition underlying YOPO's residual saliency.

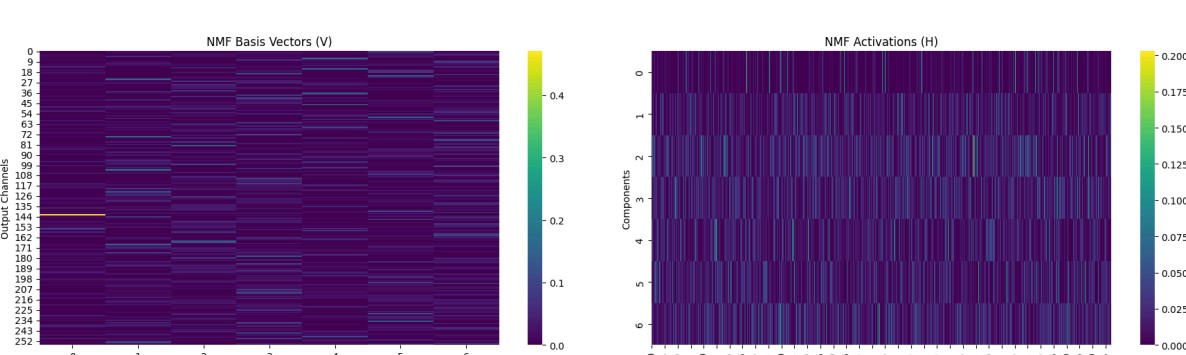

**Figure 7:** Heatmaps of NMF factors for a VGG-19 convolutional layer. **Left:** Basis matrix $V$, showing the contribution of each low-rank component to output channels. **Right:** Activation matrix $H$, illustrating component activation across flattened input features.

**Implications.**   The retained residual distribution demonstrates that YOPO does not simply enforce sparsity, but selectively removes redundant components aligned with the low-rank template. This selective preservation is consistent with the spectral analyses in the main text, where YOPO-pruned models exhibit stronger singular value concentration at high sparsities. Together, these results indicate that residual-based pruning preferentially preserves parameters that contribute structural diversity beyond the dominant low-dimensional subspace.

## I   Ablations and Stability

We analyze the sensitivity of YOPO to thresholding rules, low-rank factorization, minimal keep constraints, and initialization, and we further include sanity checks to validate the residual-based saliency ordering.

**Thresholding.**   We compare median absolute deviation (MAD) and standard deviation (STD) thresholds under both global and layer-wise sparsity. Across moderate sparsity levels, the two schemes produce nearly identical accuracy–sparsity frontiers (Fig. 9, Fig. 10). At higher sparsities ($s \geq 0.9$), residual distributions become increasingly heavy-tailed, where MAD provides slightly more stable behavior. The absolute performance gap remains small and dataset-dependent, but MAD exhibits fewer extreme deviations, motivating its adoption as the default thresholding rule.

**Low-rank factorization.**   We sweep the NMF rank $r_\ell \in \{2, 3, 4, 6, 8, 10\}$ and observe that accuracy remains largely flat across a broad intermediate range. For VGG-16 on CIFAR-100 at $s = 0.9$, both YOPO-L and YOPO-G remain stable across ranks, with YOPO-G consistently achieving higher accuracy at fixed $(r, s)$ (Fig. 4). Increasing $r_\ell$ reduces residual contrast and narrows the spread of saliency values (Fig. 8), reflecting improved low-rank approximation but weaker separation between structurally redundant and distinctive weights. Very large ranks therefore diminish the informativeness of the residual signal, while small-to-moderate ranks preserve strong contrast and stable performance.

**Minimal keep constraints.**   We study the effect of enforcing a minimum number of retained parameters per layer. The default $m_{\min} = 100$ stabilizes very deep architectures at high sparsity. Empirically, $m_{\min} \geq 40$

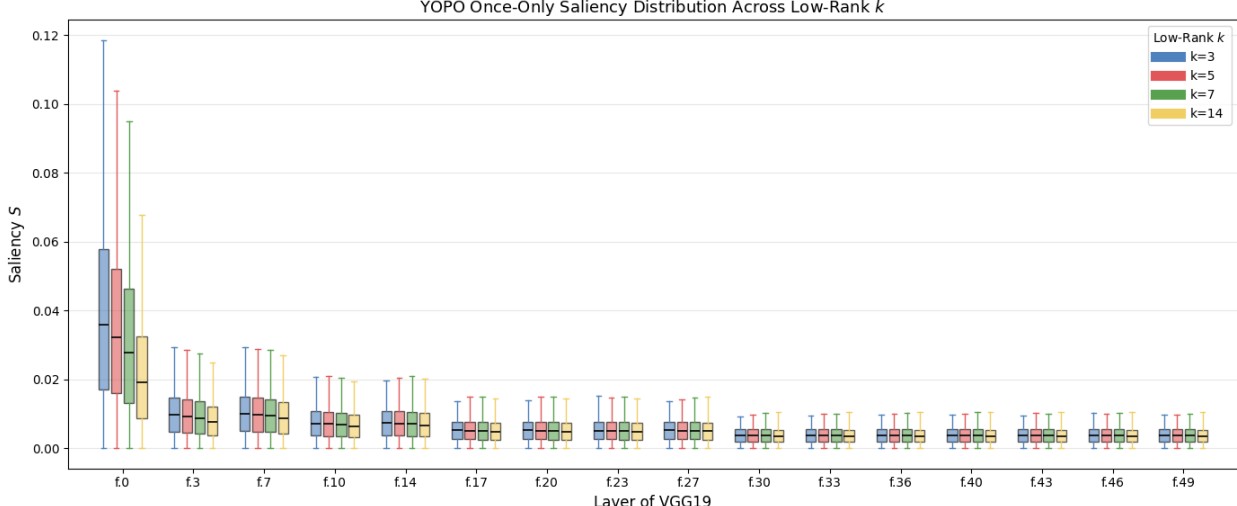

**Figure 8: Distribution of YOPO saliency scores across VGG-19 layers for different low-rank settings $k$ (with $T = 200$).** Higher ranks lower median saliency and reduce spread across layers, indicating improved low-rank approximation; later layers exhibit higher and more variable saliency.

**Table 9: Additional analysis of YOPO.** Left: robustness to initialization at global sparsity $s = 0.90$. Right: sanity check using inverse residual scores.

| Initialization | C10 | C100 |
|---|---|---|
| Normal | $93.8 \pm 0.30$ | $73.6 \pm 0.20$ |
| Xavier normal | $93.29 \pm 0.20$ | $72.09 \pm 0.30$ |
| Kaiming uniform | $93.68 \pm 0.30$ | $73.15 \pm 0.25$ |

| Method | 68.38 | 90.00 | 96.84 | 99.00 |
|---|---|---|---|---|
| YOPO (high-res.) | 59.82 | 57.88 | 50.08 | 38.80 |
| Inverse | 57.68 | 55.30 | 42.81 | 29.35 |
| Gap $\Delta$ | -2.14 | -2.58 | -7.27 | -9.45 |

Robustness to initialization at $s = 0.90$.        Accuracy (%) across sparsity levels.

slightly improves robustness at $s = 0.95$, while even $m_{\min} = 1$ prevents layer collapse at ultra-sparsity ($>$ 99%). Column-level constraints are typically unnecessary, indicating that collapse safety emerges primarily from the residual-based ordering.

**Per-channel vs. full-layer NMF.** Applying NMF independently per output channel significantly reduces preprocessing time while preserving the saliency ranking and final accuracy. Comparative experiments across backbones show negligible differences between the two formulations, suggesting that the residual structure is primarily local to channel interactions.

**Initialization robustness.** Table 9 evaluates performance across Normal, Xavier-normal, and Kaiming-uniform initializations at global sparsity $s = 0.90$. Accuracy differences remain within small margins across schemes on both CIFAR-10 and CIFAR-100. This stability aligns with the initialization-independence behavior quantified by the $I^3$ index (Table 10), where cross-seed performance variations remain minimal. The small proxy gaps $\Delta_\dagger$ indicate that masks constructed from one initialization transfer consistently to others, supporting the once-only saliency hypothesis.

Overall, these ablations demonstrate that YOPO is robust to reasonable variations in factorization rank, thresholding scheme, and initialization. Performance remains stable across seeds and sparsity budgets, while the residual-based ordering preserves its discriminative structure under both global and layer-wise pruning.

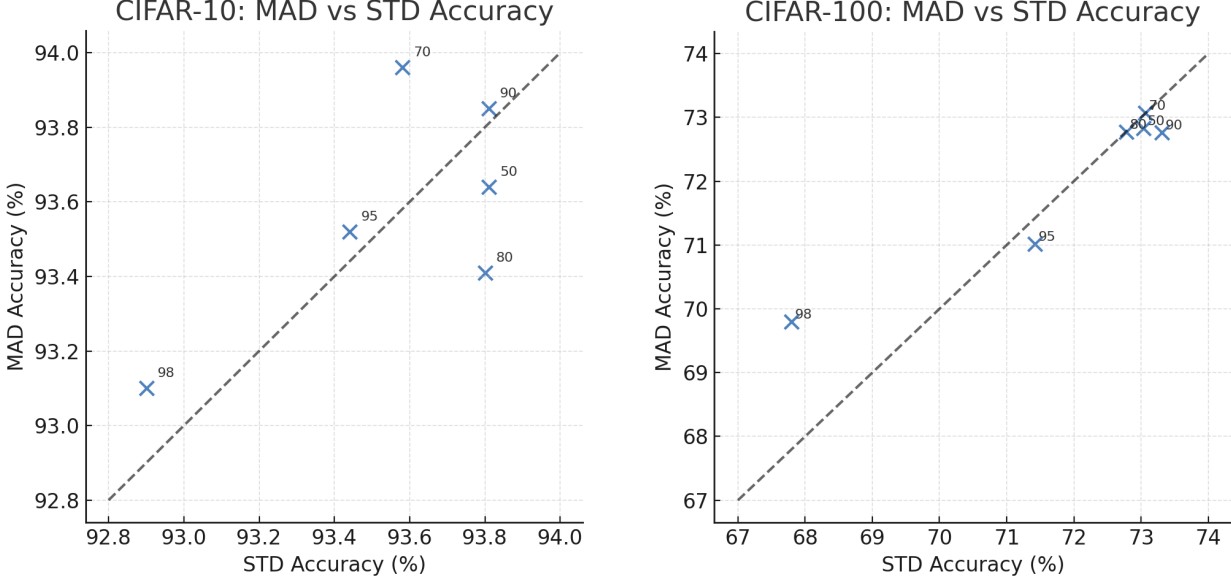

**Figure 9: MAD vs. STD pruning accuracy for YOPO-G.** Paired scatter plots comparing MAD vs. STD pruning accuracy for YOPO-G on CIFAR-10 (left) and CIFAR-100 (right). Each point corresponds to a sparsity setting, annotated by percentage. Points above the $y = x$ dashed line indicate superior MAD performance; points below favor STD.

**Table 10: Initialization-Independence Index ($I^3$) at 90% sparsity (VGG19).** $Acc_{self}$ denotes accuracy when the mask is evaluated on its own initialization. $Acc^{\dagger}_{xseed}$ is a leave-one-out proxy for cross-seed transfer. $\Delta_{\dagger}$ measures sensitivity to initialization (smaller is better).

| Dataset | Seed | $Acc_{self}$ (%) | $Acc^{\dagger}_{xseed}$ (%) | $\Delta_{\dagger}$ (%) |
|---------|------|------------------|------------------------------|-------------------------|
| CIFAR-100 | A (42) | 72.80 | 72.35 | 0.45 |
| | B (52) | 72.50 | 72.43 | 0.08 |
| | C (62) | 72.10 | 72.53 | -0.43 |
| | D (72) | 72.30 | 72.48 | -0.18 |
| | E (82) | 72.50 | 72.43 | 0.08 |
| | Mean ± Std | **72.44 ± 0.26** | | Mean$|\Delta_{\dagger}| = $ **0.24** |
| CIFAR-10 | A (42) | 93.80 | 93.50 | 0.30 |
| | B (52) | 93.40 | 93.60 | -0.20 |
| | C (62) | 93.70 | 93.53 | 0.18 |
| | D (72) | 93.10 | 93.68 | -0.58 |
| | E (82) | 93.80 | 93.50 | 0.30 |
| | Mean ± Std | **93.56 ± 0.30** | | Mean$|\Delta_{\dagger}| = $ **0.31** |

## J   Implementation Details and Reproducibility

**Environment.** PyTorch `2.x`, CUDA `11+/12+`, single A100 or V100. Random seeds fixed per run (dataloader, PyTorch, NumPy). AMP enabled for training; NMF preprocessing runs on CPU or GPU (whichever is idle).

**Baselines.** We reproduce SNIP, GraSP, and SynFlow from official or widely used reference code (matching their scoring hyperparameters) but *always* train with our shared recipe (optimizer, schedule, epochs). For

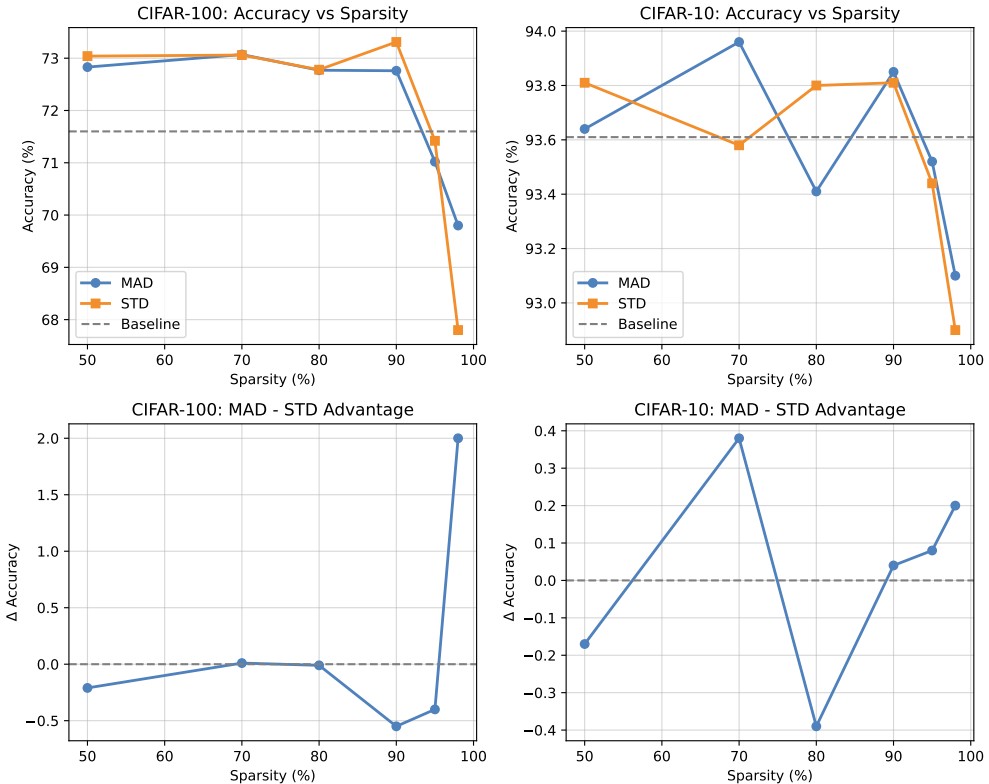

**Figure 10: Accuracy trends and $\Delta$ comparison of MAD vs. STD thresholding.** Absolute accuracy (top) and $\Delta = \text{MAD} - \text{STD}$ accuracy (bottom) across sparsity levels. CIFAR-100 (left) shows near parity with a slight STD edge at extreme sparsity, while CIFAR-10 (right) consistently favors MAD. Dashed lines mark dense baselines.

SNIP/GraSP we use a single scoring batch (size 1024 when feasible); for SynFlow we run $K{=}100$ pruning steps to the target $s$.

**Reporting.** All tables report mean ± std over 3 seeds. We release code, seeds, and masks, and provide per-layer sparsity profiles and raw logs.

# K  Experimental Details

We evaluate YOPO across multiple datasets, architectures, and sparsity regimes. All pruned models are trained from scratch after mask construction under the standard recipes listed in Table 11.

**Datasets.** We consider CIFAR-10 and CIFAR-100 (50k train / 10k test), Tiny-ImageNet (100k train / 10k validation, 200 classes, $64 \times 64$ resolution), and ImageNet-1k (1.28M train / 50k validation, $224 \times 224$ resolution). For CIFAR experiments, a temporary validation split is used only for early diagnostics; final results are obtained by retraining on the full training set and reporting performance on the official test split.

**Architectures.** We evaluate VGG-16/19 (CIFAR and ImageNet variants), ResNet-18/34/50/101, ResNet-20/56, WRN-28-10, and ConvNeXt backbones. All convolutional and fully connected layers are prunable unless otherwise stated.

**Data preprocessing and augmentation.** For CIFAR-10/100, we use per-channel normalization, random $32 \times 32$ crops with 4-pixel reflection padding, and horizontal flips ($p = 0.5$). For Tiny-ImageNet, we apply per-channel normalization, random resized crops to $64 \times 64$, and horizontal flips. For ImageNet-1k, we follow

standard resized-crop training to $224 \times 224$ with random horizontal flips and center-crop evaluation. Unless explicitly noted, no Cutout, Mixup, or label smoothing is used.

**Table 11:** Summary of the architectures, datasets, and hyperparameters used in experiments.

| Network | Dataset | Epochs | Batch | Optimizer | Momentum | LR | LR Drop, Epoch | Weight Decay |
|---------|---------|--------|-------|-----------|----------|-----|----------------|--------------|
| VGG-19 | CIFAR-100 | 160 | 128 | SGD | 0.9 | 0.1 | 10x, [60,120] | 0.0001 |
| ResNet-20 | CIFAR-10 | 160 | 128 | SGD | 0.9 | 0.1 | 10x, [60,120] | 0.0001 |
| ResNet-18 | Tiny-ImageNet | 100 | 128 | SGD | 0.9 | 0.01 | 10x, [30,60,80] | 0.0001 |

**Mask construction.** YOPO computes once-only saliency using Euclidean NMF on absolute weights $\tilde{W} = |W|$. The factorization rank $r_\ell$ is selected from $\{3, 5, 7, 11, 14\}$ (default $r_\ell = 7$), with multiplicative-update steps as specified in the implementation. Thresholding uses MAD by default, with STD considered in ablations. A minimal row-wise keep constraint prevents layer collapse. Global sparsity (YOPO-G) selects a single threshold via bisection to match the target sparsity, while layer-wise sparsity (YOPO-L) applies per-layer ratios. No data or gradients are used during saliency computation.

**Training protocol.** Optimization hyperparameters, learning-rate schedules, epochs, batch sizes, and weight decay values are architecture- and dataset-specific and summarized in Table 11. In all cases, we use SGD with momentum 0.9 unless otherwise noted. Pruned weights are frozen via gradient masking,

$$\nabla W^{(\ell)} \leftarrow \nabla W^{(\ell)} \odot M^{(\ell)},$$

ensuring that removed parameters remain inactive throughout training.

**Evaluation protocol.** We report Top-1 accuracy at matched global sparsity levels, averaging over multiple random seeds where indicated. Structural transferability is quantified using the Dataset-Transfer Index (DTI), and initialization robustness using the Initialization-Independence Index ($\text{I}^3$). Additional implementation details and ablations are provided in Appendix I.

**Hardware and software.** PyTorch (1.12+), CUDA 11+, cuDNN 8+. Experiments are run on modern NVIDIA GPUs. We log wall-clock for saliency computation (NMF), training, and calibration; see §O.

## L   YOPO Implementation Details

**Saliency construction (once-only).** For each convolutional or linear prunable layer $\ell$, we flatten to $\tilde{W}^{(\ell)} = |\text{flat}(W^{(\ell)})| \in \mathbb{R}_{\geq 0}^{o_\ell \times d_\ell}$ and compute an NMF with rank $r_\ell$:

$$\min_{V^{(\ell)} \geq 0, \ H^{(\ell)} \geq 0} \left\| \tilde{W}^{(\ell)} - V^{(\ell)} H^{(\ell)} \right\|_F^2.$$

Multiplicative updates (with $\varepsilon = 10^{-8}$ in denominators) for $T{=}200$ iterations; initialization via nonnegative SVD or Uniform$[0, 1]$. We optionally normalize $\tilde{W}^{(\ell)}$ by $\text{median}(\tilde{W}^{(\ell)}) + \epsilon$ prior to NMF and use the residual saliency $S^{(\ell)} = \left| \tilde{W}^{(\ell)} - V^{(\ell)} H^{(\ell)} \right|$.

**Thresholding and calibration.** Layerwise robust threshold

$$\tau_\ell(\alpha) = \text{median}(S^{(\ell)}) + \alpha \, \text{MAD}(S^{(\ell)}),$$

mask $M^{(\ell)}(\alpha) = \mathbf{1}[S^{(\ell)} > \tau_\ell(\alpha)]$ (strict comparison). YOPO-G uses a single $\alpha$ found by bisection to achieve target $s$ within $\pm 0.1\%$; monotonicity of $\hat{s}(\alpha)$ ensures convergence in $O(\log(1/\varepsilon))$ iterations. YOPO-L chooses $\alpha_\ell$ to meet per-layer ratios $s_\ell$.

**Survival constraints and ties.** We enforce per-row and optional per-column keeps:

$$\sum_j M_{ij}^{(\ell)} \geq m_{\min}, \qquad \sum_i M_{ij}^{(\ell)} \geq c_{\min}, \quad (m_{\min} \in \{1, 2\},\ c_{\min} \in \{0, 1\}).$$

We use strict inequality ($>$) in masking; infinitesimal jitter may be added to $S^{(\ell)}$ for deterministic tie-breaking.

**Complexity and Practical Considerations** For a prunable layer $\ell$ with reshaped absolute weights $\tilde{W}^{(\ell)} \in \mathbb{R}^{o_\ell \times d_\ell}$, one NMF run with rank $r_\ell$ and $T$ multiplicative-update iterations incurs a computational cost of

$$\tilde{\mathcal{O}}(T\, r_\ell\, o_\ell d_\ell) \quad \text{per layer.}$$

Since $r_\ell$ is small (typically 3–8) and saliency is computed only once at initialization, the overall overhead is modest relative to full training. In practice, NMF wall-clock time is one to two orders of magnitude smaller than a single training epoch for standard backbones.

For global sparsity (YOPO-G), the threshold parameter $\alpha$ is selected via bisection, which converges in $O(\log(1/\varepsilon))$ iterations to sparsity tolerance $\varepsilon$. Layer-wise sparsity (YOPO-L) avoids this global search and applies per-layer thresholds directly.

*Storage.* The saliency tensor $S^{(\ell)}$, computed once from the residual $S^{(\ell)} = |\tilde{W}^{(\ell)} - V^{(\ell)}H^{(\ell)}|$, is used only to construct the binary mask. After pruning, $S^{(\ell)}$ can be released from memory. Consequently, YOPO introduces negligible additional storage overhead beyond the mask itself[7], which matches the size required by any unstructured pruning method.

*Practical heuristics.* (i) Initialize $(V, H)$ using nonnegative SVD seeds or i.i.d. Uniform$[0, 1]$ values. (ii) Prefer MAD thresholding at high sparsity due to heavy-tailed residual distributions. (iii) Factor each output channel independently to reduce matrix dimensions and preprocessing time when needed. (iv) Optionally normalize layers prior to factorization to improve scale stability.

**Per-channel NMF (speed optimization).** A fast variant factors each output row independently (rank $r{=}1$ by default), which reduces matrix sizes and preprocessing time with similar rankings; see §I.

**MORT: Monotone Re-Thresholding for Exact Sparsity (Global or Layerwise).** Let $S^{(\ell)} = \{S_i^{(\ell)}\}$ be YOPO's once-only saliency for layer $\ell$ and $\tau_\ell(\alpha) = \text{median}(S^{(\ell)}) + \alpha\,\text{MAD}(S^{(\ell)})$ (resp. mean/STD) the robust threshold. Because $\tau_\ell(\alpha)$ is nondecreasing in $\alpha$ and $M^{(\ell)}(\alpha) = \mathbf{1}[S^{(\ell)} > \tau_\ell(\alpha)]$ is pointwise nonincreasing, the achieved sparsity $\hat{s}(\alpha)$ is *monotone* (nondecreasing) in $\alpha$ under both global and layerwise calibration. Thus any feasible budget $s^\star \in [0, 1]$ is attained to tolerance $\varepsilon$ by bisection in $\mathcal{O}(\log(1/\varepsilon))$ evaluations while preserving the *nested mask* property $M(\alpha_2) \leq M(\alpha_1)$ for $\alpha_2 > \alpha_1$.

**Observed calibration behavior (YOPO-G vs. YOPO-L).** Our experiments indicate the following qualitative patterns (consistent with prior reports that *global* budgets often preserve accuracy better than rigid per-layer quotas):

1. **Accuracy at fixed $s$.** YOPO variants across various sparsity profiles are illustrated in Figure 11a. *YOPO-G* tends to achieve *slightly higher accuracy* than *YOPO-L* at the same global target sparsity. Global calibration allows layers with intrinsically higher residual statistics to retain more parameters, improving overall capacity where the saliency indicates greater necessity.

2. **Sparsity profile across depth.** *YOPO-L* with the equal-ratio policy maintains *nearly uniform* sparsity across layers ($s_\ell \approx s$), which can be desirable for hardware constraints or fair per-layer comparison. In contrast, *YOPO-G* prunes *less* in early layers and *more* in later layers: early convolutional blocks typically exhibit *larger* residuals (edges/low-level templates are less well captured by small-rank additive factors), so their entries more often exceed $\tau_\ell(\alpha)$ and survive. This naturally recovers the widely observed

---

[7]The saliency tensor is discarded immediately after mask construction and is not stored during training.

heuristic that early layers warrant gentler pruning, without hand-crafted per-layer schedules. A heatmap of layerwise vs. global sparsity is depicted in Figure 11b.

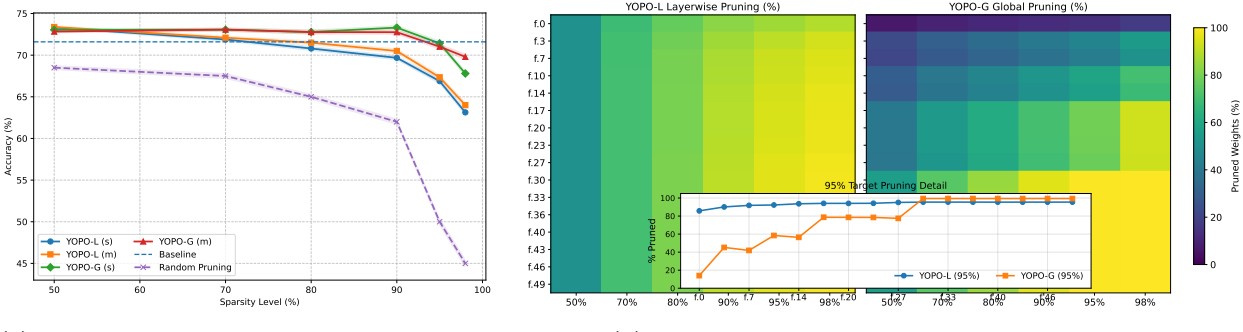

**(a)** YOPO variants vs. random pruning across sparsity.

**(b)** Layerwise vs. global sparsity allocation on VGG19.

**Figure 11: YOPO performance and sparsity allocation profiles.** (a) Across all sparsity levels, YOPO variants outperform random pruning with lower variance. (b) Heatmaps show that YOPO-L enforces near-uniform pruning, whereas YOPO-G adapts sparsity across depth—preserving early layers (14–45% pruned) and aggressively pruning deeper layers ($> 95\%$).

**Choosing between YOPO-G and YOPO-L in practice.** *Default to YOPO-G* when accuracy is the primary objective and no strict per-layer budget is mandated. *Prefer YOPO-L* when deployment requires uniform layer densities (e.g., operator fusion limits, memory partitioning) or when comparing methods under fixed per-layer sparsity. YOPO's once-only saliency $S^{(\ell)}$ supports *both* regimes without recomputation; switching between global and local calibration amounts to changing only the threshold selection rule. Numerically, Table 16 shows that this adaptive allocation yields a small but consistent accuracy edge for YOPO-G at higher budgets (e.g., $s \in \{0.9, 0.95, 0.98\}$ on CIFAR-10), while YOPO-L is competitive or best at moderate budgets where balanced compression is preferred (e.g., $s \in \{0.5, 0.7\}$).

**Percentile fallback and feasibility.** If the desired $s$ lies outside the achievable range due to survival constraints or discrete ties, we fallback to percentile-based thresholds on $S^{(\ell)}$ (global or per layer) to match the budget within tolerance. In practice this is rare because the bisection on $\alpha$ yields fine-grained control and the survival constraints affect only a negligible fraction of entries at high sparsity.

**Stability across initializations.** Because $S^{(\ell)}$ depends only on $|W^{(\ell)}|$ and a low-rank additive fit, the ordering of entries is stable across random seeds; calibration with either YOPO-G or YOPO-L therefore achieves *reproducible* sparsity and accuracy, and supports mask transfer (Sec. 6).

## M    Baselines and Fairness

**SNIP.** Connection sensitivity from one mini-batch; we use official or widely adopted implementations and calibrate sparsity with the same global/local policies and gradient masking during training. Code - https://github.com/mil-ad/snip

**GraSP.** Hessian–gradient product–based saliency; we follow common practice for batch size and Hessian approximation; sparsity calibration matches YOPO's protocols. Code - https://github.com/alecwangcq/GraSP

**SynFlow.** Iterative, data-free scoring using synaptic flow conservation; number of iterations chosen to match target $s$; we preserve positivity conventions and avoid layer collapse. Code https://github.com/ganguli-lab/Synaptic-Flow

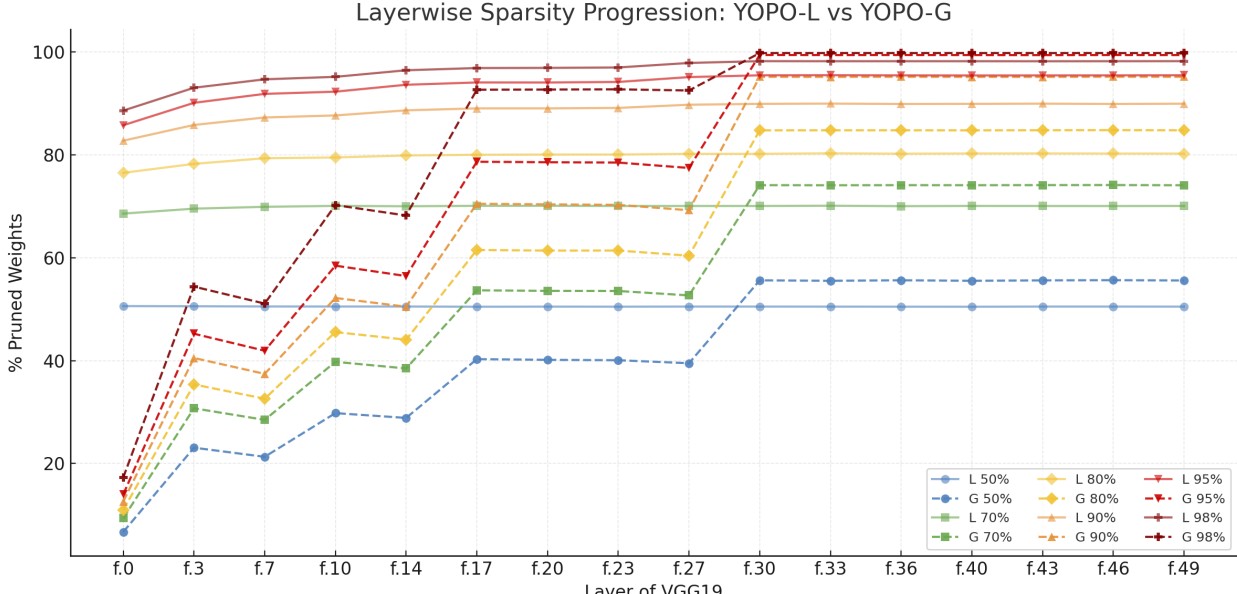

**Figure 12: Layerwise Sparsity Allocation: YOPO-L vs. YOPO-G.** Figure X presents the layerwise sparsity distribution for VGG19 under YOPO-L (solid) and YOPO-G (dashed) across target sparsities from 50% to 98%. Each curve shows the percentage of pruned weights per convolutional layer (`f.0` through `f.49`), revealing how pruning pressure is allocated through depth. YOPO-L applies nearly uniform pruning across layers (flat curves), whereas YOPO-G is adaptive: early layers are pruned much less (≈10-15% even at 95% target sparsity), while deeper layers are pruned aggressively, approaching >99% sparsity in the final convolutional blocks at high pruning ratios. This matches the expectation that early layers encode essential low-level features (edges, textures) important for generalization, while later layers contain more redundant, task-specific filters that can be pruned.

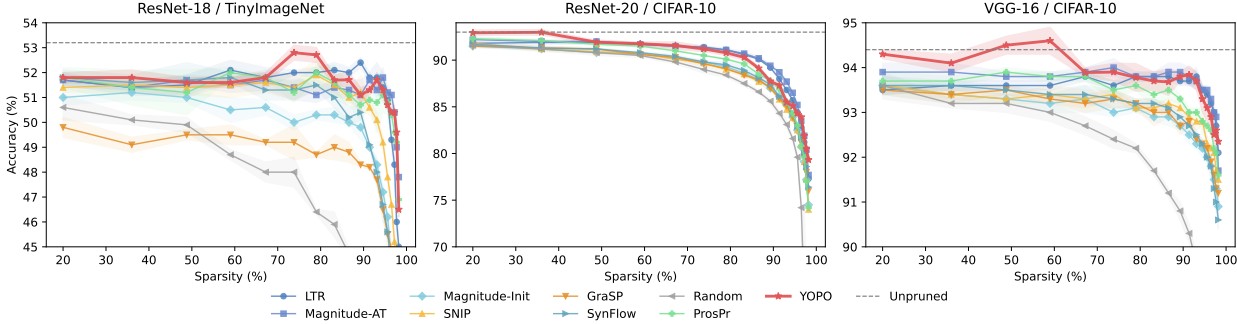

**Figure 13: Accuracy vs. Sparsity Comparison: YOPO (Data-Free) vs. Data/Gradient-Dependent Pruning at Initialization (PaIs)** tracks the best methods at moderate pruning and maintains a clear margin at high sparsity (≥ 95%), outperforming data/gradient dependent PaIs baselines (ProsPr, SNIP, GraSP, SynFlow, Magnitude@Init) under extreme compression.

**Magnitude-at-init and random.** Magnitude uses $|W|$; random-within-layer uses uniform random scores per layer with the same calibration and survival constraints. Code - publicly available.

**Training parity.** All methods share the same optimizer, schedule, epochs, augmentation, gradient masking, and evaluation metrics. Any method-specific hyperparameters are tuned within small grids that do not alter the global training recipe.

**Table 12: Comparison of Pruning at Initialization (PaI) Mechanisms.** Summary of saliency criteria, data dependence, pruning schedule, and mathematical formulation for all baselines and YOPO. "NMF" denotes nonnegative matrix factorization; "$\kappa$" the synaptic flow score; "$\nabla_W \mathcal{L}$" the gradient of training loss w.r.t. weights; "$H$" the Hessian; "$K$" the NTK. Data-Free ($\star$) methods require no mini-batch at pruning time.

| Method | Saliency Criterion | Data-Free? | Schedule | Mathematical Formulation |
|---|---|---|---|---|
| **YOPO (Ours)** | NMF Residual | $\star$ Yes | One-shot | $S_{ij} = \left|\tilde{W}_{ij} - (VH)_{ij}\right|$, where $\tilde{W} = |W|$, $VH \approx \tilde{W}$ (NMF rank-$r$). Ranking fixed once at init; re-thresholded for any budget. |
| Magnitude (Ramanujan et al., 2020) | Absolute weight | $\star$ Yes | One-shot | $S_{ij} = |W_{ij}|$. Rank by absolute weight value; threshold at desired sparsity. |
| Random | Uniform random | $\star$ Yes | One-shot | $S_{ij} \sim \text{Uniform}[0,1]$ i.i.d.; equivalent to random mask. |
| SNIP (Lee et al., 2019) | Connection sensitivity | No | One-shot | $S_{ij} = \left|\frac{\partial \mathcal{L}(x,y)}{\partial W_{ij}}\right|$; evaluated on one mini-batch $(x,y)$ before training. |
| GraSP (Wang et al., 2020) | Gradient–Hessian product | No | One-shot | $S_{ij} \propto -[H\nabla_W \mathcal{L}]_{ij} \cdot W_{ij}$; preserves gradient flow (Taylor expansion of loss change). Requires a mini-batch and Hessian approximation. |
| ProsPr (Alizadeh et al., 2022) | Meta-gradient | No | One-shot | Score via unrolled gradient of sparse training objective; requires labeled data and inner-loop meta-optimization. |
| SynFlow (Tanaka et al., 2020) | Synaptic flow conservation | $\star$ Yes | Iterative | $S_{ij}^{(t)} = \left[\frac{\partial \mathcal{R}(\theta)}{\partial W_{ij}} \cdot W_{ij}\right]$ with $\mathcal{R}(\theta) = \mathbf{1}^\top \left(\prod_\ell |W^{(\ell)}|\right)\mathbf{1}$; iterative layerwise pruning ($T$ steps) to conserve flow and avoid layer collapse. |
| PHEW (Patil & Dovrolis, 2021) | Random-walk path weight | $\star$ Yes | One-shot | Score each weight by the number of random walks that traverse it in the computational graph; data-free topological heuristic. |
| NPB (Pham et al., 2023) | Node–path balance | $\star$ Yes | Iterative | Minimise structural imbalance $\Delta(G) = |\log N_{\text{eff}} - \log P_{\text{eff}}|$ w.r.t. binary mask; iterative discrete optimization over node/path counts. |
| DPaI (Xiang et al., 2025) | Differentiable node–path balance | $\star$ Yes | Iterative | Relaxes NPB to continuous mask via sigmoid; gradient-based structural program minimising imbalance $\Delta(G)$ with straight-through estimator. |

---

**Algorithm 2 MORT**: Monotone Re-Thresholding for Exact Sparsity (Global or Layerwise)

---

**Require:** Scores $\{S^{(\ell)}\}_{\ell=1}^{L}$; target budget $s^{\star} \in [0,1]$; MODE $\in \{\text{GLOBAL}, \text{LAYERWISE}\}$; ROBUST $\in \{\text{MAD}, \text{STD}\}$; tolerance $\varepsilon > 0$; optional MIN-KEEP

**Ensure:** Mask $M = \{M^{(\ell)}\}$ with realized sparsity $\hat{s}(\alpha^{\star}) \approx s^{\star}$

1: **if** MODE=GLOBAL **then** $S_{\text{all}} \leftarrow \bigcup_{\ell} S^{(\ell)}$ **end if**
2: **for** $\ell = 1$ **to** $L$ **do**          ▷ precompute stats
3:      $(\text{loc}_\ell, \text{scale}_\ell) \leftarrow \textbf{stat}(S^{(\ell)}, \text{ROBUST})$         ▷ &MAD or &STD
4: **end for**
5: **if** MODE=GLOBAL **then**
6:      $(\text{loc}_{\text{all}}, \text{scale}_{\text{all}}) \leftarrow \textbf{stat}(S_{\text{all}}, \text{ROBUST})$
7: **end if**
8: **function** ACHIEVEDSPARSITY($\alpha$)
9:      **if** MODE=GLOBAL **then**
10:         $\tau \leftarrow \text{loc}_{\text{all}} + \alpha \, \text{scale}_{\text{all}}$
11:         **for** $\ell = 1$ **to** $L$ **do**
12:            $M^{(\ell)}(\alpha) \leftarrow \mathbf{1}[\, S^{(\ell)} > \tau \,]$
13:         **end for**
14:      **else**              ▷ LAYERWISE
15:         **for** $\ell = 1$ **to** $L$ **do**
16:            $\tau_\ell \leftarrow \text{loc}_\ell + \alpha \, \text{scale}_\ell$
17:            $M^{(\ell)}(\alpha) \leftarrow \mathbf{1}[\, S^{(\ell)} > \tau_\ell \,]$
18:         **end for**
19:      **end if**
20:      **if** MIN-KEEP **then**          ▷ collapse avoidance
21:         enforce $\geq 1$ survivor per output row/channel in each $M^{(\ell)}(\alpha)$
22:      **end if**
23:      **return** $\hat{s}(\alpha) \leftarrow \dfrac{|\{w : M_w(\alpha) = 0\}|}{|w|}$
24: **end function**
25: **function** BRACKET($f$)          ▷ find $[\alpha_{\text{lo}}, \alpha_{\text{hi}}]$ with $f(\alpha_{\text{lo}}) \leq 0 \leq f(\alpha_{\text{hi}})$
26:      $\alpha_{\text{lo}} \leftarrow 0, \ \alpha_{\text{hi}} \leftarrow 1$
27:      **while** $f(\alpha_{\text{hi}}) < 0$ **do**
28:         $\alpha_{\text{hi}} \leftarrow 2 \cdot \alpha_{\text{hi}}$
29:      **end while**
30:      **return** $(\alpha_{\text{lo}}, \alpha_{\text{hi}})$
31: **end function**
     **Bisection on $\alpha$ (monotone target):**
32: $f(\alpha) \leftarrow$ ACHIEVEDSPARSITY($\alpha$) $- s^{\star}$
33: $(\alpha_{\text{lo}}, \alpha_{\text{hi}}) \leftarrow$ BRACKET($f$)
34: **while** $\alpha_{\text{hi}} - \alpha_{\text{lo}} > \varepsilon$ **do**
35:      $\alpha \leftarrow 0.5(\alpha_{\text{lo}} + \alpha_{\text{hi}})$
36:      **if** ACHIEVEDSPARSITY($\alpha$) $< s^{\star}$ **then**
37:         $\alpha_{\text{lo}} \leftarrow \alpha$
38:      **else**
39:         $\alpha_{\text{hi}} \leftarrow \alpha$
40:      **end if**
41: **end while**
42: $\alpha^{\star} \leftarrow \alpha_{\text{hi}};$    recompute $M = \{M^{(\ell)}(\alpha^{\star})\}$ via ACHIEVEDSPARSITY
43: **return** $M$

---

## N    Transfer Metrics and Protocol

**Once-only masks across budgets.**    For any $p \in [0, 1]$, we obtain $m_p$ by re-thresholding the fixed saliency $S(\theta_0)$ (YOPO-G/L) without recomputation.

**Dataset transfer.**    We compute $S(\theta_0)$ once per backbone, then evaluate training on CIFAR-10 and CIFAR-100 by reusing the same mask family (or the same $S$ with a new threshold) across datasets. For resusing the same mask on TinyImageNet we resize the input.

**Metrics.**    Primary: Top-1 accuracy (%), #Params. Transfer/stability:

$$\text{DTI}(D_{\text{src}} \to D_{\text{tgt}}, p) = \mathcal{R}\big(p; D_{\text{tgt}} \mid \mathcal{M}_{\text{src}}(p)\big) - \mathcal{R}\big(p; D_{\text{tgt}} \mid \mathcal{M}_{\text{tgt}}(p)\big),$$
$$\text{I}^3(D, p) = \tfrac{1}{S} \textstyle\sum_{s=1}^{S} \big(\mathcal{R}(p; D \mid \mathcal{M}^{(s)}(p)) - \mathcal{R}(p; D \mid \mathcal{M}^{(\bar{s})}(p))\big).$$

Lower is better; values near 0 indicate successful transfer and initialization robustness.

**Budgets and seeds.**    We evaluate $p \in \{0.5, 0.7, 0.8, 0.9, 0.95\}$ and seeds $\{1, 2, 3\}$.

**FLOPs and parameters.**    We count multiply-adds for convolutions and fully connected layers; Batch-Norm/activations are excluded. Structured FLOPs after kernel/channel removal are *not* reported in the main paper (future work).

## O    Additional Results Tables

**Table 13:** Correlation between NMF residual magnitude and NTK-based parameter influence (sum of squared gradients) across convolutional layers of the YOPO-pruned model. Positive correlation indicates alignment between structural uniqueness and functional importance.

| Metric | Average | Median |
|---|---|---|
| Pearson $r$ | 0.2794 | 0.2382 |
| Spearman $\rho$ | 0.4208 | 0.4145 |

**Table 14:** Singular value decay statistics for a representative convolutional layer (`layers.12.conv.weight`) at ~95% sparsity. YOPO exhibits the steepest decay and fastest variance concentration, supporting the low-rank template hypothesis.

| Model | $\sigma_1$ | Decay Rate | # SVs for 90% Var. | Effective Rank Trend |
|---|---|---|---|---|
| Dense | ~1.87 | Slowest | >2000 | Highest |
| SynFlow | ~0.82 | Moderate | 1500–2000 | Reduced |
| Random | ~0.42 | Moderate | 1500–2000 | Reduced |
| **YOPO** | ~0.35 | **Steepest** | ~1000 | **Lowest** |

## P    Applicability to Large Language Models

The low-rank residual saliency underlying YOPO depends only on the weight matrix of a linear projection layer and a nonnegative factorization thereof. It requires no architectural assumptions about convolutions, making it directly applicable to the linear layers of transformer-based large language models (LLMs).

**Setup.**    We evaluate YOPO on **LLaMA-2-7B** (Touvron et al., 2023) and **Phi-2 (2.7B)** (Li et al., 2023) at unstructured sparsity levels of 20%, 25%, and 30%. For each target MLP weight matrix (feed-forward linear blocks), we apply NMF with a uniform rank of $r$=7. To construct the global binary pruning mask across all

**Table 15: Mask transfer across sparsity levels.** Top-1 accuracy (%) on CIFAR-10/100 for VGG19 and WRN28-10. Note: Saliency is computed once per backbone and re-thresholded to each target sparsity without rescoring.

| Model | #Params | Dataset | Unpruned | Sparsity level (% zeros) | | | | | |
|---|---|---|---|---|---|---|---|---|---|
| | | | | 50% | 70% | 80% | 90% | 95% | 98% |
| VGG19 | 20M | CIFAR-10 | 93.61 | 93.60 | 93.90 | 93.40 | 93.80 | 93.50 | 93.10 |
| | | CIFAR-100 | 71.60 | 72.83 | 73.07 | 72.77 | 72.76 | 71.02 | 69.80 |
| WRN28-10 | 37M | CIFAR-10 | 96.40 | 97.26 | 96.85 | 96.79 | 96.83 | 95.38 | 94.81 |
| | | CIFAR-100 | 80.70 | 80.79 | 81.06 | 80.77 | 80.75 | 79.07 | 77.89 |

**Table 16: ResNet-56 on CIFAR-10/100 at multiple sparsity levels.** Top-1 accuracy (%) as mean $\pm$ std over seeds. Best and second-best per column (within each dataset block) are **bold** and *italic*, respectively. Dense baselines: CIFAR-10 = 93.50%, CIFAR-100 = 72.54%.

| | ResNet-56 + CIFAR-10 | | | | |
|---|---|---|---|---|---|
| Sparsity | 50% | 70% | 90% | 95% | 98% |
| LTH (Frankle & Carbin, 2019) | $92.67 \pm 0.25$ | $91.88 \pm 0.35$ | $89.78 \pm 0.35$ | $88.05 \pm 0.50$ | $83.85 \pm 0.55$ |
| LTH Iter-5 (Frankle & Carbin, 2019) | $92.68 \pm 0.39$ | $92.50 \pm 0.15$ | $90.24 \pm 0.27$ | $88.10 \pm 0.36$ | $83.91 \pm 0.15$ |
| EB (You et al., 2019) | $92.76 \pm 0.21$ | $91.61 \pm 0.60$ | $89.50 \pm 0.60$ | $88.00 \pm 0.38$ | $83.74 \pm 0.35$ |
| Scratch | $92.49 \pm 0.35$ | $92.14 \pm 0.27$ | $89.89 \pm 0.12$ | $87.41 \pm 0.31$ | $82.71 \pm 0.40$ |
| GraSP (Wang et al., 2020) | $91.82 \pm 0.25$ | $90.85 \pm 0.28$ | $89.93 \pm 0.20$ | $88.42 \pm 0.30$ | $84.16 \pm 0.15$ |
| RST (Bai et al., 2022) | $92.34 \pm 0.12$ | $92.27 \pm 0.24$ | $90.41 \pm 0.05$ | $88.24 \pm 0.08$ | $83.77 \pm 0.47$ |
| RST Iter-5 (Bai et al., 2022) | *93.41 $\pm$ 0.16* | $92.67 \pm 0.02$ | *90.43 $\pm$ 0.21* | $88.40 \pm 0.14$ | $83.97 \pm 0.09$ |
| YOPO-G (M)[8] | $93.10 \pm 0.16$ | *92.85 $\pm$ 0.03* | **91.83 $\pm$ 0.06** | **89.52 $\pm$ 0.14** | **87.70 $\pm$ 0.15** |
| YOPO-L (S)[8] | **93.44 $\pm$ 0.06** | **93.09 $\pm$ 0.21** | $90.15 \pm 0.21$ | *89.37 $\pm$ 0.03* | *85.82 $\pm$ 0.16* |
| | ResNet-56 + CIFAR-100 | | | | |
| Sparsity | 50% | 70% | 90% | 95% | 98% |
| LTH (Frankle & Carbin, 2019) | $69.95 \pm 0.47$ | $68.24 \pm 0.60$ | $65.66 \pm 0.47$ | $60.97 \pm 0.30$ | $52.77 \pm 0.44$ |
| LTH Iter-5 (Frankle & Carbin, 2019) | $70.57 \pm 0.15$ | $69.54 \pm 0.46$ | $64.84 \pm 0.11$ | $60.45 \pm 0.61$ | $53.83 \pm 0.09$ |
| EB (You et al., 2019) | $70.27 \pm 0.59$ | $69.16 \pm 0.36$ | $64.01 \pm 0.42$ | $60.09 \pm 0.33$ | $53.14 \pm 1.04$ |
| Scratch | $70.96 \pm 0.25$ | $68.39 \pm 0.35$ | $64.62 \pm 0.52$ | $59.93 \pm 0.24$ | $50.80 \pm 0.55$ |
| GraSP (Wang et al., 2020) | $67.98 \pm 0.15$ | $67.38 \pm 0.25$ | $64.21 \pm 0.25$ | $59.39 \pm 0.25$ | $45.01 \pm 0.25$ |
| RST (Bai et al., 2022) | $71.13 \pm 0.48$ | $69.85 \pm 0.23$ | $66.17 \pm 0.18$ | $61.66 \pm 0.37$ | $54.11 \pm 0.37$ |
| RST Iter-5 (Bai et al., 2022) | $71.39 \pm 0.34$ | $70.48 \pm 0.19$ | $65.65 \pm 0.15$ | *61.71 $\pm$ 0.36* | *54.46 $\pm$ 0.32* |
| YOPO-G (M)[9] | *71.58 $\pm$ 0.16* | *70.90 $\pm$ 0.09* | **68.61 $\pm$ 0.19** | $61.35 \pm 0.12$ | **59.07 $\pm$ 0.26** |
| YOPO-L (S)[9] | **71.85 $\pm$ 0.21** | **71.08 $\pm$ 0.13** | *68.53 $\pm$ 0.06* | **61.81 $\pm$ 0.26** | $52.74 \pm 0.18$ |

MLP layers without rescoring, we perform a binary search over the NMF residual scores to find the global MAD threshold corresponding to the target sparsity budget. Attention projection matrices (query, key, value, output), embeddings, and layer normalizations are kept dense. We compare against **SliceGPT** (Ashkboos et al., 2024) and **PruneNet** (Sengupta et al., 2025), two representative LLM compression baselines, at their reported 20% compression level. All models are evaluated in zero-shot across five commonsense reasoning benchmarks using the `lm-evaluation-harness` framework.

**Zero-shot reasoning accuracy.** Table 19 and Table 20 report the Top-1 accuracy on five reasoning benchmarks for LLaMA-2-7B and Phi-2, respectively. On LLaMA-2-7B, YOPO achieves near-lossless performance at 20% sparsity (68.87% average accuracy compared to 69.00% for dense) and retains over 99.2% of the dense average accuracy even at 30% sparsity. On Phi-2, YOPO slightly improves over dense zero-shot accuracy at 20% and 25% sparsity (71.67% and 71.56% vs. 71.47% dense). In both architectures, YOPO

**Table 17: Mask transfer across datasets at fixed sparsity budgets.** Note: The transferability is measured in DTI, Definition 1. Columns show per-dataset DTI and two summaries: mean DTI and worst absolute DTI.

| Model | #Params | $p$ | C10 | C100 | TinyIN | Mean | Worst \|DTI\| |
|-------|---------|-----|------|------|--------|------|------------|
| VGG16 | 14M | 0.90 | $-0.10$ | $-0.05$ | $-0.09$ | $-0.08$ | 0.10 |
| VGG19 | 20M | 0.90 | $-0.12$ | 0.08 | 0.04 | 0.00 | 0.12 |
| RN18 | 11M | 0.90 | 0.01 | $-0.02$ | 0.12 | 0.04 | 0.12 |
| RN34 | 21M | 0.90 | 0.14 | $-0.06$ | 0.01 | 0.03 | 0.14 |
| VGG16 | 14M | 0.95 | 0.07 | $-0.40$ | $-0.21$ | $-0.18$ | 0.40 |
| VGG19 | 20M | 0.95 | $-0.06$ | 0.10 | $-0.18$ | $-0.05$ | 0.18 |
| RN18 | 11M | 0.95 | 0.11 | 0.02 | $-0.11$ | 0.01 | 0.11 |
| RN34 | 21M | 0.95 | 0.02 | 0.03 | $-0.07$ | $-0.01$ | 0.07 |

substantially outperforms the structured pruning baseline methods at 20% compression without requiring any calibration data or gradient computation.

**Inference throughput.** Table 21 reports wall-clock generation throughput (tokens/sec) measured on a single NVIDIA A100-40GB GPU with batch size 8 generating up to 128 new tokens, using standard `transformers` inference without custom sparse kernels. As expected for unstructured pruning without hardware-aligned sparse matrix acceleration, throughput gains are modest (1.3% to 1.6%): the dominant computational bottlenecks at inference time are memory bandwidth and attention KV-cache lookup rather than dense multiply-accumulate operations in the MLP blocks. Achieving proportional latency reductions in standard deployment requires physical slicing or hardware-specific sparse kernels.

**Discussion.** These results demonstrate that YOPO's data-free NMF residual saliency generalizes naturally to transformer LLMs with no architectural modifications. The near-lossless accuracy at 20% to 30% sparsity confirms that low-rank additive templates are present at random initialization in LLM weight matrices, and that parameters deviating from these templates remain functionally important regardless of architecture. The wall-clock throughput behavior is intrinsic to unstructured pruning on current hardware, consistent with the discussion in Section D. Extending YOPO to structured slicing (e.g., scoring and physically removing entire MLP intermediate neurons based on aggregated NMF residuals) represents a natural and promising direction for achieving physical model compression and direct latency reductions.

**Table 18:** Test accuracy (%) across YOPO vs data/gradient based pruning methods at different sparsity levels.

| Sparsity (%) | 20 | 36 | 48.8 | 59 | 67.2 | 73.8 | 79 | 83.2 | 86.6 | 89.3 | 91.4 | 93.1 | 94.5 | 95.6 | 96.5 | 97.2 | 97.7 | 98.2 |
|---|---|---|---|---|---|---|---|---|---|---|---|---|---|---|---|---|---|---|
| **VGG-16 on CIFAR-10** | | | | | | | | | | | | | | | | | | |
| LTR after Training | 93.5 ± 0.2 | 93.6 ± 0.1 | 93.6 ± 0.1 | 93.6 ± 0.1 | 93.8 ± 0.1 | 93.6 ± 0.1 | 93.8 ± 0.1 | 93.8 ± 0.1 | 93.8 ± 0.1 | 93.7 ± 0.1 | 93.7 ± 0.1 | 93.8 ± 0.2 | 93.5 ± 0.1 | 93.4 ± 0.1 | 93.2 ± 0.1 | 93.0 ± 0.2 | 92.7 ± 0.1 | 92.1 ± 0.4 |
| Magnitude after Training | 93.9 ± 0.2 | 93.9 ± 0.2 | 93.8 ± 0.1 | 93.8 ± 0.1 | 93.9 ± 0.1 | 94.0 ± 0.2 | 93.8 ± 0.1 | 93.8 ± 0.1 | 93.9 ± 0.1 | 93.9 ± 0.2 | 93.8 ± 0.2 | 93.7 ± 0.2 | 93.5 ± 0.1 | 93.5 ± 0.1 | 93.3 ± 0.2 | 93.0 ± 0.1 | 92.9 ± 0.1 | 91.7 ± 0.8 |
| Magnitude at initialization | 93.6 ± 0.2 | 93.4 ± 0.2 | 93.3 ± 0.1 | 93.2 ± 0.2 | 93.3 ± 0.3 | 93.0 ± 0.1 | 93.1 ± 0.1 | 92.9 ± 0.1 | 92.9 ± 0.2 | 92.7 ± 0.1 | 92.5 ± 0.2 | 92.3 ± 0.1 | 92.2 ± 0.2 | 92.0 ± 0.1 | 91.8 ± 0.2 | 91.5 ± 0.1 | 91.3 ± 0.3 | 90.9 ± 0.2 |
| SNIP | 93.6 ± 0.1 | 93.4 ± 0.1 | 93.3 ± 0.1 | 93.4 ± 0.2 | 93.3 ± 0.2 | 93.4 ± 0.1 | 93.1 ± 0.1 | 93.1 ± 0.1 | 93.2 ± 0.1 | 93.1 ± 0.1 | 92.9 ± 0.1 | 92.8 ± 0.2 | 92.8 ± 0.1 | 92.3 ± 0.2 | 92.2 ± 0.1 | 92.1 ± 0.1 | 91.7 ± 0.1 | 91.5 ± 0.1 |
| GraSP | 93.5 ± 0.1 | 93.4 ± 0.2 | 93.5 ± 0.1 | 93.3 ± 0.1 | 93.2 ± 0.2 | 93.3 ± 0.2 | 93.2 ± 0.1 | 93.0 ± 0.1 | 93.0 ± 0.1 | 93.0 ± 0.1 | 93.0 ± 0.1 | 92.4 ± 0.2 | 92.3 ± 0.1 | 92.2 ± 0.1 | 91.9 ± 0.1 | 91.6 ± 0.1 | 91.3 ± 0.0 | 91.2 ± 0.2 |
| SynFlow | 93.6 ± 0.2 | 93.6 ± 0.1 | 93.5 ± 0.1 | 93.4 ± 0.1 | 93.4 ± 0.2 | 93.3 ± 0.2 | 93.2 ± 0.1 | 93.2 ± 0.1 | 93.1 ± 0.1 | 92.9 ± 0.1 | 92.7 ± 0.2 | 92.5 ± 0.1 | 92.3 ± 0.1 | 92.0 ± 0.1 | 91.8 ± 0.3 | 91.3 ± 0.1 | 91.0 ± 0.2 | 90.6 ± 0.2 |
| Random | 93.6 ± 0.3 | 93.2 ± 0.1 | 93.2 ± 0.2 | 93.0 ± 0.2 | 92.7 ± 0.2 | 92.4 ± 0.2 | 92.2 ± 0.1 | 91.7 ± 0.1 | 91.2 ± 0.1 | 90.8 ± 0.2 | 90.3 ± 0.2 | 89.6 ± 0.2 | 88.8 ± 0.2 | 88.3 ± 0.4 | 87.6 ± 0.1 | 86.4 ± 0.2 | 86.0 ± 0.4 | 84.5 ± 0.4 |
| ProsPr | 93.7 ± 0.2 | 93.7 ± 0.1 | 93.9 ± 0.1 | 93.8 ± 0.1 | 93.8 ± 0.1 | 93.5 ± 0.2 | 93.6 ± 0.1 | 93.5 ± 0.2 | 93.5 ± 0.2 | 93.3 ± 0.1 | 93.0 ± 0.1 | 93.0 ± 0.1 | 92.8 ± 0.3 | 92.7 ± 0.1 | 92.6 ± 0.1 | 92.2 ± 0.1 | 92.1 ± 0.2 | 91.6 ± 0.4 |
| YOPO | 94.3 ± 0.1 | 94.1 ± 0.2 | 94.5 ± 0.2 | 94.6 ± 0.3 | 93.89 ± 0.1 | 93.9 ± 0.2 | 93.78 ± 0.3 | 93.7 ± 0.2 | 93.68 ± 0.3 | 93.8 ± 0.1 | 93.84 ± 0.2 | 93.7 ± 0.2 | 93.3 ± 0.3 | 93.1 ± 0.3 | 92.9 ± 0.2 | 92.5 ± 0.2 | 92.6 ± 0.3 | 92.35 ± 0.2 |
| **ResNet-20 on CIFAR-10** | | | | | | | | | | | | | | | | | | |
| LTR after Training | 91.8 ± 0.2 | 91.9 ± 0.2 | 91.9 ± 0.2 | 91.7 ± 0.2 | 91.5 ± 0.1 | 91.4 ± 0.1 | 91.1 ± 0.1 | 90.6 ± 0.1 | 90.1 ± 0.1 | 89.2 ± 0.1 | 88.0 ± 0.2 | 86.8 ± 0.2 | 85.7 ± 0.1 | 84.4 ± 0.2 | 82.8 ± 0.1 | 81.2 ± 0.3 | 79.4 ± 0.3 | 77.3 ± 0.5 |
| Magnitude after Training | 92.2 ± 0.3 | 92.0 ± 0.2 | 92.0 ± 0.2 | 91.7 ± 0.1 | 91.5 ± 0.2 | 91.3 ± 0.2 | 91.1 ± 0.2 | 90.7 ± 0.2 | 90.2 ± 0.2 | 89.4 ± 0.2 | 88.7 ± 0.2 | 87.7 ± 0.2 | 86.5 ± 0.2 | 85.2 ± 0.3 | 83.5 ± 0.3 | 81.9 ± 0.3 | 80.4 ± 0.2 | 77.7 ± 0.4 |
| Magnitude at initialization | 91.5 ± 0.2 | 91.2 ± 0.1 | 90.8 ± 0.1 | 90.7 ± 0.2 | 90.2 ± 0.1 | 89.8 ± 0.2 | 89.3 ± 0.2 | 88.6 ± 0.2 | 87.9 ± 0.3 | 87.0 ± 0.3 | 86.1 ± 0.2 | 85.2 ± 0.4 | 83.9 ± 0.2 | 82.5 ± 0.4 | 80.7 ± 0.5 | 79.1 ± 0.4 | 77.2 ± 0.4 | 74.5 ± 0.7 |
| SNIP | 91.8 ± 0.2 | 91.2 ± 0.3 | 90.9 ± 0.1 | 90.7 ± 0.1 | 90.1 ± 0.2 | 89.7 ± 0.3 | 89.0 ± 0.2 | 88.5 ± 0.3 | 87.7 ± 0.2 | 87.2 ± 0.4 | 85.8 ± 0.1 | 84.7 ± 0.3 | 83.8 ± 0.3 | 82.5 ± 0.4 | 80.9 ± 0.2 | 79.1 ± 0.2 | 77.3 ± 0.2 | 74.0 ± 0.5 |
| GraSP | 91.5 ± 0.1 | 91.3 ± 0.2 | 91.2 ± 0.1 | 90.6 ± 0.2 | 90.3 ± 0.2 | 89.6 ± 0.1 | 89.1 ± 0.2 | 88.4 ± 0.2 | 87.9 ± 0.1 | 87.0 ± 0.2 | 85.9 ± 0.1 | 85.1 ± 0.4 | 83.9 ± 0.4 | 82.8 ± 0.2 | 81.2 ± 0.2 | 79.7 ± 0.3 | 78.0 ± 0.3 | 76.0 ± 0.5 |
| SynFlow | 91.7 ± 0.1 | 91.3 ± 0.2 | 91.2 ± 0.1 | 90.8 ± 0.1 | 90.4 ± 0.2 | 89.8 ± 0.1 | 89.5 ± 0.3 | 88.9 ± 0.4 | 88.1 ± 0.1 | 87.4 ± 0.5 | 86.1 ± 0.2 | 85.4 ± 0.2 | 84.3 ± 0.2 | 82.9 ± 0.2 | 81.7 ± 0.2 | 80.0 ± 0.3 | 78.6 ± 0.4 | 76.4 ± 0.4 |
| Random | 91.6 ± 0.2 | 91.2 ± 0.2 | 90.8 ± 0.3 | 90.5 ± 0.2 | 89.8 ± 0.2 | 89.0 ± 0.4 | 88.4 ± 0.2 | 87.5 ± 0.3 | 86.6 ± 0.2 | 85.6 ± 0.3 | 84.3 ± 0.4 | 83.1 ± 0.4 | 81.6 ± 0.3 | 79.6 ± 0.4 | 74.2 ± 0.4 | 64.7 ± 9.7 | 56.9 ± 8.5 | 43.7 ± 12.5 |
| ProsPr | 92.3 ± 0.1 | 92.1 ± 0.0 | 91.7 ± 0.2 | 91.5 ± 0.1 | 91.0 ± 0.2 | 90.5 ± 0.0 | 90.1 ± 0.1 | 89.6 ± 0.2 | 88.5 ± 0.5 | 87.8 ± 0.1 | 86.9 ± 0.3 | 85.5 ± 0.6 | 84.3 ± 0.2 | 83.0 ± 0.9 | 80.8 ± 0.5 | 79.6 ± 0.7 | 77.0 ± 0.8 | 74.2 ± 0.3 |
| YOPO | 92.93 ± 0.3 | 92.98 ± 0.3 | 91.92 ± 0.2 | 91.76 ± 0.2 | 91.57 ± 0.4 | 91.2 ± 0.3 | 90.79 ± 0.3 | 90.3 ± 0.3 | 89.13 ± 0.2 | 87.73 ± 0.2 | 87.3 ± 0.4 | 85.5 ± 0.3 | 85.1 ± 0.2 | 84.3 ± 0.3 | 83.9 ± 0.3 | 81.9 ± 0.2 | 80.5 ± 0.4 | 79.33 ± 0.5 |
| **ResNet-18 on TinyImageNet** | | | | | | | | | | | | | | | | | | |
| LTR after Training | 51.7 ± 0.2 | 51.4 ± 0.1 | 51.5 ± 0.4 | 52.1 ± 0.4 | 51.8 ± 0.4 | 52.0 ± 0.1 | 52.0 ± 0.2 | 52.0 ± 0.2 | 52.1 ± 0.3 | 52.0 ± 0.2 | 52.4 ± 0.2 | 51.8 ± 0.6 | 51.8 ± 0.6 | 51.4 ± 0.4 | 50.9 ± 0.2 | 49.3 ± 0.7 | 48.3 ± 0.7 | 46.0 ± 0.3 |
| Magnitude after Training | 51.7 ± 0.3 | 51.4 ± 0.1 | 51.7 ± 0.2 | 51.5 ± 0.3 | 51.7 ± 0.4 | 51.4 ± 0.5 | 51.1 ± 0.3 | 51.4 ± 0.3 | 51.3 ± 0.4 | 51.1 ± 0.6 | 51.7 ± 0.3 | 51.3 ± 0.3 | 51.8 ± 0.4 | 51.2 ± 0.3 | 51.1 ± 0.2 | 50.4 ± 0.2 | 49.0 ± 0.2 | 47.8 ± 0.5 |
| Magnitude at Initialization | 51.0 ± 0.3 | 51.2 ± 0.3 | 51.0 ± 0.2 | 50.5 ± 0.5 | 50.6 ± 0.3 | 50.0 ± 0.3 | 50.3 ± 0.3 | 50.0 ± 0.1 | 50.0 ± 0.1 | 49.8 ± 0.5 | 49.0 ± 0.1 | 48.3 ± 0.3 | 47.2 ± 0.2 | 46.2 ± 0.2 | 44.4 ± 0.5 | 42.2 ± 0.1 | 40.8 ± 0.4 | 38.1 ± 0.6 |
| SNIP | 51.4 ± 0.2 | 51.5 ± 0.3 | 51.4 ± 0.3 | 51.5 ± 0.5 | 51.6 ± 0.4 | 51.4 ± 0.5 | 51.9 ± 0.6 | 51.5 ± 0.3 | 51.0 ± 0.2 | 51.2 ± 0.7 | 50.6 ± 0.3 | 50.1 ± 0.3 | 49.2 ± 0.3 | 47.8 ± 0.2 | 46.7 ± 0.1 | 45.2 ± 0.4 | 44.5 ± 0.3 | 42.3 ± 0.3 |
| GraSP | 49.8 ± 0.4 | 49.1 ± 0.3 | 49.5 ± 0.2 | 49.5 ± 0.4 | 49.2 ± 0.1 | 49.2 ± 0.7 | 48.7 ± 0.1 | 49.0 ± 0.5 | 48.8 ± 0.4 | 48.3 ± 0.1 | 48.2 ± 0.1 | 47.7 ± 0.2 | 46.5 ± 0.1 | 45.5 ± 0.7 | 44.9 ± 0.2 | 44.1 ± 0.1 | 42.9 ± 0.5 | 41.0 ± 0.1 |
| SynFlow | 51.8 ± 0.3 | 51.6 ± 0.3 | 51.7 ± 0.7 | 51.8 ± 0.2 | 51.3 ± 0.4 | 51.3 ± 0.4 | 51.5 ± 0.2 | 51.0 ± 0.4 | 50.2 ± 0.4 | 50.4 ± 0.3 | 49.1 ± 0.0 | 48.0 ± 0.5 | 46.7 ± 0.7 | 45.6 ± 0.0 | 44.0 ± 0.2 | 42.2 ± 0.3 | 40.0 ± 0.1 | 38.2 ± 0.5 |
| Random | 50.6 ± 0.5 | 50.1 ± 0.2 | 49.9 ± 0.3 | 48.7 ± 0.2 | 48.0 ± 0.4 | 48.0 ± 0.6 | 46.4 ± 0.1 | 45.9 ± 0.4 | 44.7 ± 0.2 | 43.6 ± 0.3 | 42.7 ± 0.2 | 41.4 ± 0.4 | 40.2 ± 0.2 | 37.2 ± 0.2 | 36.2 ± 0.7 | 34.0 ± 0.4 | 32.2 ± 0.5 | 30.0 ± 0.3 |
| ProsPr | 51.8 ± 0.4 | 51.4 ± 0.7 | 51.2 ± 0.9 | 52.0 ± 0.2 | 51.8 ± 0.1 | 51.2 ± 0.4 | 52.0 ± 0.3 | 51.6 ± 0.7 | 51.1 ± 0.4 | 50.7 ± 0.6 | 50.9 ± 0.3 | 50.8 ± 1.2 | 51.1 ± 0.7 | 50.8 ± 0.5 | 50.3 ± 0.8 | 49.6 ± 0.6 | 49.2 ± 0.2 | 46.9 ± 0.7 |
| YOPO (64x64 settings) | 51.8 ± 0.3 | 51.8 ± 0.3 | 51.6 ± 0.2 | 51.6 ± 0.2 | 51.8 ± 0.2 | 52.8 ± 0.4 | 52.71 ± 0.4 | 51.71 ± 0.2 | 51.71 ± 0.3 | 51.1 ± 0.3 | 51.3 ± 0.3 | 51.7 ± 0.3 | 51.3 ± 0.3 | 50.7 ± 0.4 | 50.4 ± 0.5 | 50.4 ± 0.4 | 49.6 ± 0.5 | 46.5 ± 0.3 |
| YOPO (224x224 settings) | 65.1 ± 0.3 | 64.5 ± 0.3 | 63.51 ± 0.4 | 63.53 ± 0.2 | 63.4 ± 0.3 | 62.9 ± 0.3 | 62.81 ± 0.4 | 62.51 ± 0.3 | 62.41 ± 0.5 | 61.78 ± 0.4 | 61.55 ± 0.5 | 59.55 ± 0.4 | 58.1 ± 0.4 | 58.27 ± 0.3 | 57.27 ± 0.5 | 54.01 ± 0.3 | 52.01 ± 0.5 | 50.01 ± 0.4 |

**Table 19: Zero-shot reasoning accuracy (%) on LLaMA-2-7B.** YOPO is applied once at initialization using data-free NMF residuals; masks are reused across all tasks. Baseline results are taken from their respective papers at 20% compression.

| Method | Sparsity | PIQA | HellaSwag | ARC-e | ARC-c | WinoGrande | Avg. |
|---|---|---|---|---|---|---|---|
| Dense | 0% | 79.11 | 75.99 | 74.58 | 46.25 | 69.06 | 69.00 |
| SliceGPT (Ashkboos et al., 2024) | 20% | 69.42 | 59.04 | 59.76 | 37.54 | 65.11 | 58.17 |
| PruneNet (Sengupta et al., 2025) | 20% | 75.30 | 66.43 | 63.80 | 37.29 | 65.51 | 61.67 |
| YOPO (Ours) | 20% | **78.78** | **75.46** | **74.49** | **46.42** | **69.22** | **68.87** |
| YOPO (Ours) | 25% | 78.94 | 75.29 | 74.33 | 46.50 | 69.46 | 68.90 |
| YOPO (Ours) | 30% | 78.89 | 74.91 | 73.99 | 45.90 | 68.67 | 68.47 |

**Table 20: Zero-shot reasoning accuracy (%) on Phi-2 (2.7B).** YOPO maintains or slightly surpasses dense baseline accuracy across all evaluated sparsity budgets without data or retraining.

| Method | Sparsity | PIQA | HellaSwag | ARC-e | ARC-c | WinoGrande | Avg. |
|---|---|---|---|---|---|---|---|
| Dense | 0% | 74.37 | 73.07 | 81.31 | 52.82 | 75.77 | 71.47 |
| PruneNet (Sengupta et al., 2025) | 20% | 74.37 | 65.53 | 74.71 | 47.53 | 70.80 | 66.59 |
| YOPO (Ours) | 20% | **78.94** | **72.68** | **78.24** | **52.65** | **75.85** | **71.67** |
| YOPO (Ours) | 25% | **79.38** | **72.23** | 77.90 | 52.22 | **76.09** | **71.56** |
| YOPO (Ours) | 30% | 78.78 | 71.74 | 78.24 | 51.11 | 75.14 | 71.00 |

**Table 21: Inference throughput (tokens/sec) with YOPO at 20% sparsity.** Measured on an A100-40GB GPU (batch size 8, generating 128 tokens). Modest wall-clock gains confirm the known limitation of unstructured pruning without hardware-specific sparse kernels.

| Model | Dense | YOPO 20% Sparse | Relative Gain |
|---|---|---|---|
| LLaMA-2-7B | 258.47 tok/s | 261.79 tok/s | +1.3% |
| Phi-2 (2.7B) | 311.25 tok/s | 316.15 tok/s | +1.6% |

