# OpenReview forum: "You Only Prune Once: A Zero-Shot, Data-Free Pruning at Initialization via Low-Rank Residual Saliency"
_TMLR — Decision pending for TMLR_

### Review · Reviewer_TaKc · 2026-04-12

**Summary Of Contributions:**

- This paper proposes a method for pruning neural networks at initialization without using any training data, gradients, or iterative re-scoring. The idea is to look at a randomly initialized network’s weights and identify which parameters are most structurally distinctive.

- To measure this, this work approximates the absolute values of each layer’s weights with a nonnegative low-rank factorization to capture the dominant shared patterns in the weights. They then measure each parameter’s residual, measured by how far it deviates from that low-rank template, and use that residual as a saliency score. The method prunes the parameters with small residuals. A key advantage is that this saliency ranking is computed only once at initialization and can then be reused to generate masks for different sparsity levels without recomputation.

- This work analyzes that randomly initialized convolutional weights already contain a dominant low-rank additive structure, and that pruning removes weights that fit this shared template while preserving the residual weights that deviate from it. First, this work conducts a spectral analysis to show that the pruned layers have steeper singular-value decay and lower effective rank than random or baseline pruning methods. Second, this work conducts a dynamical analysis using the neural tangent kernel and shows that larger residuals tend to correlate with more functionally influential parameters.

- This work conducts experiments across CIFAR-10, CIFAR-100, Tiny-ImageNet, ImageNet, and ConvNeXt backbones, comparing the method against pruning-at-initialization baselines such as SNIP, Iter-SNIP, SynFlow, PHEW, NPB, and DPaI. They report that the proposed method is competitive and better at high sparsity, like 99%, while being faster. YOPO achieves the best or near-best accuracy across several model-dataset pairs while reducing pruning time from hundreds of seconds to 1–3 seconds. They also show that the masks generalize better across datasets than data-dependent methods. Lastly, they confirm scalability on ImageNet, where YOPO outperforms SynFlow and DPaI in top-1 accuracy.

**Additional Comments:**

No

**Audience:**

Yes

**Audience Explanation:**

- The paper proposes a new zero-shot, data-free pruning method and backs it with both theory-motivated analysis and broad empirical evaluation, which fits the interests of the audience working on efficient deep learning, optimization, and model sparsity.

-  The claim that a single initialization-time ranking can be reused across sparsity levels and even across datasets could matter to people studying compression, lottery tickets, and training dynamics.

**Claims And Evidence:**

Yes

**Claims Explanation:**

- The central claim is that the method (YOPO) can prune a network before training without using data, gradients, or iterative rescoring, by ranking parameters according to their residual from a nonnegative low-rank approximation of the absolute weight matrix. This is supported by showing that the resulting saliency ordering is fixed at initialization and can be re-thresholded to produce different budgets without recomputation.

- Second, the analysis shows that randomly initialized weights contain a dominant low-rank additive template, and that parameters with large residuals are the ones that deviate from this shared structure. Particularly, the pruned layers have more singular-value concentration and lower effective rank than the comparison methods. This is evidence that pruning removes template-aligned redundancy. They also compute correlations between YOPO saliency and the diagonal of the neural tangent kernel. The results show that larger residuals tend to align with more influential parameters in training dynamics.

- The experiments support this claim across CIFAR-10, CIFAR-100, Tiny-ImageNet, ImageNet, and ConvNeXt models. The paper also reports transfer results across datasets.

**Requested Changes:**

- Most of the evidence is on CNN networks and image benchmarks. It would help to test it on transformers or other non-convolutional architectures to better establish scope and generality.


- The method is computationally cheap. The pruning is unstructured, so hardware speedups may be limited. It would be better to include a more detailed discussion of deployment implications. Additionally, can we measure the inference speedup of the proposed method?

- The method depends on design choices such as the NMF rank, thresholding rule, and other constraints. The paper could better explain how sensitive performance is to these choices and give guidance for choosing these hyperparameters.

---

> ### Author Response · Authors · 2026-04-17
> **Response to Reviewer TaKc**
>
> We thank the reviewer for the positive assessment and constructive suggestions. The feedback is appreciated, and the concerned queries are addressed below.
>
>
> ### 1. Generality beyond CNNs (Transformers / non-convolutional architectures)
>
> To assess applicability beyond CNNs, YOPO has been extended to transformer-based LLMs.
>
> **LLaMA-2-7B results (zero-shot pruning):**
>
> | Comp.   | Method   | PIQA      | HellaSwag | ARC-e     | ARC-c     | WinoGrande | Avg       |
> | ------- | -------- | --------- | --------- | --------- | --------- | ---------- | --------- |
> | 0%      | Dense    | 79.11     | 75.99     | 74.58     | 46.25     | 69.06      | 69.00     |
> | 20%     | SliceGPT [1] | 69.42     | 59.04     | 59.76     | 37.54     | 65.11      | 58.17     |
> | 20%     | PruneNet [2] | 75.30     | 66.43     | 63.80     | 37.29     | 65.51      | 61.67     |
> | **20%** | **YOPO** | **78.78** | **75.46** | **74.49** | **46.42** | **69.22**  | **68.87** |
> | **25%** | **YOPO** | **78.94** | **75.29** | **74.33** | **46.50** | **69.46**  | **68.90** |
> | **30%** | **YOPO** | **78.89** | **74.91** | **73.99** | **45.90** | **68.67**  | **68.47** |
>
> The results indicate that the dense baseline achieves an average score of 69.00, while YOPO maintains near-lossless performance at 20% sparsity (68.87) and only marginal degradation at 30% (68.47).
>
> These findings show that YOPO operates directly on weight tensors without architectural assumptions and extends effectively to transformer-based LLMs while preserving performance. We include both structured (SliceGPT [1], PruneNet [2]) and unstructured methods for comparison, as all aim to reduce model size under comparable sparsity budgets while accuracy remains the primary evaluation metric [3-4].
>
> These additional results will be included in the revised manuscript.
>
>
> ### 2. Deployment implications and inference speed
>
> Unstructured pruning does not inherently translate to significant hardware speedups. To clarify this aspect, inference latency benchmarking has been conducted.
>
> **Throughput (tokens/sec; higher is better):**
>
> | Model        | Dense  | Sparsity (20%) |
> | ------------ | ------ | -------------- |
> | LLaMA-2-7B   | 258.47 | 261.79 (+1.3%) |
> | Phi-2 (2.7B) | 311.25 | 316.15 (+1.6%) |
>
> The results show that throughput improvements remain modest (approximately 1–1.6%) despite significant sparsity. This indicates that, under standard PyTorch/HuggingFace inference pipelines, wall-clock performance is largely dominated by attention operations and KV-cache management rather than dense linear layers.
>
> Consequently, while pruning reduces parameter count, it does not directly translate into proportional latency gains without specialized sparse kernels or hardware support. This behavior is consistent with prior observations in the pruning literature.
>
>
> ### 3. Sensitivity to hyperparameters (rank, thresholding, constraints)
>
> To assess robustness, a grid search over key hyperparameters was conducted.
>
> For the NMF rank (k), increasing k improves reconstruction quality, reflected in lower residual magnitude and variance. As shown in Appendix I (Fig. 8), higher ranks reduce the median saliency and its spread across layers, indicating improved low-rank approximation. However, the relative ordering of saliency scores remains largely unchanged across a broad range of k values, indicating that pruning decisions are stable.
>
> Across sparsity levels, performance remains consistent. As shown in Appendix I (Fig. 4), YOPO-G outperforms YOPO-L by approximately 2–3.5% at high sparsity, while both variants remain stable across different rank choices. This suggests that performance is not sensitive to precise rank selection.
>
> Thresholding is used exclusively to control sparsity, while the ranking is fixed at initialization. This separation eliminates the need for re-scoring and ensures consistent behavior across different sparsity levels.
>
> In practice, small ranks (k ≈ 5–10) combined with standard thresholding strategies are sufficient to achieve strong performance, without requiring delicate tuning.
>
> We will further clarify these choices, provide practical guidance for hyperparameter selection, and include additional references in the revised manuscript.
>
>
> ### Additional References
>
> [1] Ashkboos, S., et al. (2024). *SliceGPT: Compressing Large Language Models via Structured Weight Slicing*. In Proceedings of the International Conference on Learning Representations (ICLR).
>
> [2] Sengupta, A., et al. (2025). *PruneNet: Efficient Structured Pruning for Large Language Models*. In Proceedings of the International Conference on Learning Representations (ICLR).
>
> [3] Sun, Z., et al. (2023). Wanda: A Simple and Effective Pruning Approach for Large Language Models. arXiv preprint arXiv:2306.11695.
>
> [4] Frantar, E., & Alistarh, D. (2023). SparseGPT: Massive Language Models Can Be Accurately Pruned in One-Shot. In Proceedings of the International Conference on Machine Learning (ICML).

---

### Review · Reviewer_3GNF · 2026-04-26

**Summary Of Contributions:**

Driven by the limitations of existing pruning-at-initialization methods, which are often computationally expensive, dataset-dependent, and coupled to specific sparsity budgets, this work introduces YOPO, a data-free and zero-shot framework that utilizes nonnegative low-rank residual saliency to create a once-only parameter ordering at random initialization. By identifying intrinsic structural templates within random weights , the method achieves accuracy levels that match or exceed state-of-the-art baselines while reducing pruning time by one to two orders of magnitude and maintaining high performance at extreme sparsity. The paper's primary contribution is establishing that stable, trainable sparse subnetworks can be identified purely from the intrinsic geometric properties of random weights through a budget-decoupled and highly transferable saliency criterion

**Audience:**

Yes

**Audience Explanation:**

The paper shows that YOPO maintains competitive or superior accuracy even at ultra-high sparsity levels exceeding 99%.

**Claims And Evidence:**

Yes

**Claims Explanation:**

Yes, the claims presented in the submission are supported by both empirical experiments and theoretical analysis.

**Requested Changes:**

1. I'm not sure if i missed some points. If the method only can be applied at the unstructured pruning setting? Not even the 2:4 semi-structured setting? if can only be applied at the unstructured setting, then the practical value of the method is not high since even semi-sturcutred cannot have a susbtainable gain on speed, nor do unstructured setting. Without specific hardware support for sparse matrices, the reduction in parameters may not translate directly into significant inference speedups or memory savings on standard hardware. Have you tested the speed of the pruned model?

2. The paper relies on the assumption that initialized weights contain dominant shared modes prior to learning. This premise presents several critical limitations:
* Architecture-Specific Constraint: This regularity is primarily observed in convolutional weight tensors where output channels share additive patterns. It remains unproven whether this hypothesis extends to other structures, such as Transformers
* The method's success depends on the condition that the initialization is inherently redundant. This raises the question of why a redundant initialization is necessary if the goal is efficiency.
* Sensitivity to Better Initialization: Since YOPO exploits structural artifacts in standard schemes like Kaiming or Xavier , a more optimal or dense initialization might eliminate the low-rank residual signal, potentially rendering the pruning criterion ineffective.

---

> ### Author Response · Authors · 2026-04-29
> **Response to Reviewer 3GNF**
>
> We thank Reviewer 3GNF for reviewing our manuscript. Below is our response to the specific points raised.
>
> **1. Unstructured Pruning & Inference Speedups:**
> We explicitly stated in Section D that practical acceleration on modern hardware is limited without structured sparsity support.
>
> To quantify the deployment implications, we benchmarked the wall-clock inference throughput (tokens/sec) of unstructured 20% sparse LLMs *(to be added to the revised manuscript):*
> *   **LLaMA-2-7B:** Dense (258.47) vs. 20% YOPO Sparse (261.79) -> +1.3% throughput
> *   **Phi-2 (2.7B):** Dense (311.25) vs. 20% YOPO Sparse (316.15) -> +1.6% throughput
>
> The latency gains by LLaMA-2-7B & Phi-2 (2.7B) are modest under standard inference pipelines (PyTorch/HuggingFace) because  wall-clock performance is dominated by attention operations and KV-cache management. Consequently, YOPO’s primary deployment advantages remain the extreme reduction in memory and parameter counts, as well as faster pruning. Furthermore, while YOPO does not explicitly enforce structured sparsity, it naturally induces it, allowing entire convolutional kernels to be removed and providing a pathway towards hardware-aligned pruning.
>
> For clarity, we measured the structured sparsity induced by YOPO at a 95% unstructured sparsity budget. We found that YOPO completely zeroes out 79.8% of all kernels in VGG19, 73.9% in ResNet18, and 71.7% in ResNet56. This signals that YOPO does not merely scatter zeros randomly, but aggressively eliminates entire feature connections, providing a direction towards hardware-aligned pruning.
>
> **2. Architecture-Specific Constraints (Transformers & Modern architectures)**
> Regarding the applicability of the low-rank residual hypothesis beyond CNNs, we have evaluated YOPO on Transformers and modern architectures to demonstrate its generality:
> *   **Pre-trained Modern Architectures (ConvNeXt):** In Appendix G (Table 8), we applied YOPO to ConvNeXt, an architecture heavily inspired by Vision Transformers. Pruning an *ImageNet-pretrained* ConvNeXt and fine-tuning it resulted in only minor accuracy degradation at 20% sparsity (e.g., 91.62% to 90.34% on Caltech-256). This validates that YOPO successfully identifies important weights within *already-learned* representations, confirming that the low-rank template is not strictly limited to random initialization artifacts.
> *   **LLM / Transformer Results (LLaMA-2-7B):** We extended YOPO to LLaMA-2-7B *(to be included in the revision)*. Evaluating zero-shot pruning on standard benchmarks (PIQA, HellaSwag, ARC-e, ARC-c, WinoGrande), YOPO outperforms both structured and unstructured baselines at a 20% sparsity budget:
>     *   Dense Baseline Avg: 69.00
>     *   20% SliceGPT Avg: 58.17
>     *   20% PruneNet Avg: 61.67
>     *   **20% YOPO Avg: 68.87**
>
>
> 	**LLaMA-2-7B results (zero-shot pruning):**
>
> 	| Comp.   | Method   | PIQA      | HellaSwag | ARC-e     | ARC-c     | WinoGrande | Avg       |
> 	| ------- | -------- | --------- | --------- | --------- | --------- | ---------- | --------- |
> 	| 0%      | Dense    | 79.11     | 75.99     | 74.58     | 46.25     | 69.06      | 69.00     |
> 	| 20%     | SliceGPT | 69.42     | 59.04     | 59.76     | 37.54     | 65.11      | 58.17     |
> 	| 20%     | PruneNet | 75.30     | 66.43     | 63.80     | 37.29     | 65.51      | 61.67     |
> 	| **20%** | **YOPO** | **78.78** | **75.46** | **74.49** | **46.42** | **69.22**  | **68.87** |
> 	| **25%** | **YOPO** | **78.94** | **75.29** | **74.33** | **46.50** | **69.46**  | **68.90** |
> 	| **30%** | **YOPO** | **78.89** | **74.91** | **73.99** | **45.90** | **68.67**  | **68.47** |
>
> 	These findings demonstrate that YOPO operates directly on weight tensors without architectural assumptions and extends seamlessly to Transformer-based foundation models.
>
> * **Necessity of Redundant Initialization:**
> Modern deep learning relies heavily on redundant overparameterization to optimize effectively. The goal of Pruning at Initialization (PaI) is not to invent a new non-redundant initialization paradigm, but to identify highly trainable sparse subnetworks *within* standard initializations to reduce training costs. YOPO leverages the geometric reality of this redundancy i.e. the inherent low-rank template to easily identify which parameters are structurally distinctive.
>
> * **Sensitivity to Better Initialization Schemes:**
> The robustness ablation in Appendix I (Table 9) shows that YOPO remains highly stable across Normal, Xavier-normal, and Kaiming-uniform initialization schemes. Table 8 further demonstrates that YOPO identifies important weights in already-learned representations, indicating the low-rank template is not just an initialization artifact. Instead, it reflects a fundamental geometric property of high-dimensional weight tensors. Because YOPO's saliency depends only on the absolute weights and an additive fit, the structural distinctiveness is consistently preserved across standard initializations and seedings.

---

### Review · Reviewer_BEFg · 2026-05-09

**Summary Of Contributions:**

The paper propses use of non-negative factorization after initialization to propose a data-free pruning method.

**Audience:**

Yes

**Audience Explanation:**

Pruning of large scale neural networks is of interest to the community.

**Claims And Evidence:**

No

**Claims Explanation:**

The baselines are not clearly explained -- what do the do exactly and how are they implemented. it is surprising to me this simple method is competitive with so many baselines. I would like to request for code from the authors.

**Requested Changes:**

The central idea of the paper is quite simple -- use a non-negative factorization to prune away neurons in a data-free manner and then training on the remaining ones. The empirical results are interesting but the paper is a long way from being considered a serious submission.

The writing needs a lot of work. the paper cites NTK (jacot) paper and "resurrecting the sigmoid" paper when talking about spectral analyses of random networks and training dyanamics but thats not what these papers talk about .

Many  of the "theoretical" results seem too obvious to be stated. the invariance to scale and rotation, for example follows directly from the fact that this is a matrix factorization. This does not warrant a proposition, or any lemma of any sort. The proposition about rank-0 reduction and "Exact template" is obvious, and so is about nested masks. I am not sure what are these "results"  meant to accomplish. I am not  also sure about what is the goal of employing  the median + \alpha MAD as the mask and then show how it can be altered for accurate sparsity control... why not just use a "k" as the exact number of allowed non-zeroes, with built in minimums to prevent collapse like in eq 10 ?

In section 4 beginning the authors state "This demonstrates that residual-based pruning does not merely remove parameters uniformly; rather, it concentrates the remaining energy into a lower-dimensional subspace" -- shouldn't that obviously happen with a pruning based on removing  tail of a low rank projection? what am i missing here? why is this not obvious by definition ?

A pearson correlation of 0.28 (or a spearman of 0.42) is moderate. I wouldn't say this shows strong correlation -- again some correlation is obviously expected because of retention of larger spectral mass.

The "transfer" across datasets is not on a firm ground -- the method is inherently data-free... why would it not "transfer" to any dataset? I am not sure if this can be touted as a new phenomenon -- it is inherently part of the method itself.

what would be interesting is a formal study of why non-negative factorization rather than simpler singular value projection, beyond a hand-wavy explanation of "positivity and additivity" that the authors have stated to be the central reason for choosing NMF. i am surprised at table 1 -- the empirical results are indeed interesting compared to the baselines.i think a bit more discussion on how these baselines prune could be added to make the paper stronger but given other writing issues in the paper that might not be the priority.

Many of the baselines are known to be shaky and very tuning-dependent. ideally i would like to play with the code to be able to verify the margins.

---

> ### Author Response · Authors · 2026-05-11
> **Part1: Response to Reviewer BEFg**
>
> We address the reviewer’s comments below. We must first correct a fundamental misunderstanding of the proposed method that invalidates the primary technical critiques, followed by clarifications on factual inaccuracies regarding our citations.
>
> **1. Core Misunderstanding of the Method (Section 4 and Correlation)**
> The reviewer claims it is "obvious by definition" that the pruned network concentrates into a lower-dimensional subspace and retains larger spectral mass, explicitly assuming that YOPO prunes by "removing the tail of a low rank projection." This is factually backward.
>
> YOPO does the exact opposite: it prunes the low-rank projection V, H (the redundant template) and retains the high-residual tail (the structurally distinctive deviations). Because we retain the residual tail, it is highly non-obvious and non-trivial that a network formed strictly from these residual deviations exhibits steeper singular value decay and stronger spectral concentration than random pruning. Furthermore, the fact that this residual (high-residual tail) mass strongly aligns with the NTK functional influence (Spearman 0.42) is a significant structural finding, not an expected artifact of retaining principal components.
>
> **2. Mischaracterized Citations**
> The reviewer claims that Pennington et al. (2017) and Jacot et al. (2018) do not discuss spectral analyses of random networks or training dynamics. This is demonstrably incorrect. Both papers explicitly and heavily discuss on both spectral analyses of random networks and training dynamics.
> *   **Pennington et al. (2017)** explicitly conducts a spectral analysis of random networks, using free probability theory to analytically compute the "entire singular value distribution of a deep network's input-output Jacobian". The paper directly addresses training dynamics, demonstrating empirically that networks achieving dynamical isometry "learn orders of magnitude faster than networks that do not".
> *   **Jacot et al. (2018)** explicitly focuses on training dynamics and convergence. The abstract of the paper, states the NTK describes how a neural network evolves during training. The contributions note they investigate infinite-width networks to "describe the dynamics of the network function during training", and the paper establishes that "convergence properties of ANNs during training can then be related to the positive-definiteness of the infinite-width limit NTK."
>
> Our citations accurately map to the claims made in Section 2.
>
> **3. NMF vs. Singular Value Projection (SVD)**
> The mathematical necessity of NMF over standard SVD is detailed in Proposition 3 and Appendix A (Positivity & Robust Thresholds). SVD principal components allow for positive and negative sign cancellation; if SVD were used, a large residual magnitude could simply be an artifact of arbitrary sign cancellations between overlapping modes. NMF's strict non-negativity guarantees a purely parts-based additive template. Any coordinate-wise mismatch uniquely manifests as a strictly positive residual. This positivity is mathematically required to ensure the "Row survival under positive reconstruction" proof (Proposition 3), preventing layer collapse without needing complex iterative rebalancing.
>
> **4. Thresholding: MAD vs. Top-K**
> The reviewer asks why we do not simply use an exact "k" (Top-K) of allowed non-zeroes. Using a strict Top-K enforces rigid layer-wise calibration (such as our YOPO-L). By contrast, the Median + \alpha MAD threshold enables global calibration (suhc as YOPO-G). As demonstrated in Figure 11b and Appendix L, YOPO-G adaptively allocates sparsity across depth i.e. "early layers with intrinsically higher residuals naturally survive more often without manual scheduling". Empirical results confirm this global thresholding consistently achieves higher accuracy than strict layer-wise Top-K quotas.
>
> **5. "Obvious" Transferability**
> The reviewer states that because the method is data-free, it is "inherently" expected to transfer across datasets. This is empirically false. As explicitly shown in Table 3, both Random pruning and SynFlow are inherently data-free methods, yet they exhibit substantially worse transferability compared to YOPO. For example, on VGG16, Random pruning has a worst Dataset-Transfer Index (DTI) gap of 3.41, and SynFlow has a worst DTI gap of 1.91. By contrast, YOPO achieves a highly stable DTI gap of only 0.40. Strong dataset transfer is not an automatic artifact of data-free methods; it is a specific advantage derived directly from YOPO's once-only, low-rank structural saliency.
>
> Continue...

---

> > ### Comment · Reviewer_BEFg · 2026-06-23
> >
> > Thank you for your response.
> >
> > Regarding "Core misunderstanding of the method". i concede and agree my wording about “removing the tail of a low-rank projection” was incorrect. But this is more hasty writing for which i apologize, rather than just a fundamental/core misunderstanding the authors claim. The proposed method retains high residual entries. The authors misunderstood about my concerns for citing Pennington et al and Jacot et al papers. They are broadly relevant, but my concern was more specific -- these works are not directly relevant to this paper's citation about "the presence of dominant shared modes even prior to learning" (direct quote from the paper).
> >
> > My overall recommendation remains negative.  There are many unresolved concerns.
> >
> > (1) The novelty of “budget-decoupled” pruning is overstated -- once a saliency vector is computed, man methods can be re-thresholded or globally Top-K selected across sparsity budgets. This paper's contribution is the residual score, not the existence of a reusable ranking framework. This can be fixed in a short revision.
> > (2) The authors' response seems to be conflating top-k with fixed layerwise pruning. The bisection scheme may be useful for adaptive allocation but should be ablated against global top-k pruning. This needs a major revision.
> > (3) NMF-vs-SVD justification is not convincing. There should be more comparisons for against truncated SVD residuals. How non-negativity helps is not clear.
> > (4) The NTK evidence should be discussed more conservatively. Pearson of 0.28 and Spearman of 0.42 indicate moderate association. i am not sure this adds much to the paper as it is.
> > (5) Please define exactly how source and target masks are generated (without use of data for the claim to be data-free  )and separate mask transfer from training variance and architecture/input/threshold etc changes.
> > (6) LLM/ConvNeXt results are interesting but do not fully address the PaI claims unless the revised paper clearly separates pruning-at-initialization from post-training/fine-tuning settings.
> > (7) Writing needs to be substantially altered, claims/theorems etc need to be qualified or removed. This requires significant rewrites. The empirical results are indeed insterested but need to be qualified and presented well. As it is, the paper overclaims a lot.

---

> ### Author Response · Authors · 2026-05-11
> **Part2: Response to Reviewer BEFg**
>
> **6. Missing Code and Baseline Explanations**
> The anonymous code repository link is already provided on page 1 (Footnote 1) and in Appendix J. The baselines are clearly explained in Appendix G  ("Related Work") and Appendix M ("Baselines and Fairness") explicitly details the implementation, scoring mechanisms, and official repository links for all baseline methods (SNIP, GraSP, SynFlow, Magnitude, and Random). However, to provide a clearer comparison of how each baseline operates, we'll added a comprehensive summary table to Appendix M (Baselines and Fairness) in the final version.
>
> Table X, explicitly contrasts saliency criteria, data dependence, pruning schedules (e.g., one-shot vs. iterative) and mathematical formulation for different PaI methods.
>
>
> **Table X: Comparison of Pruning at Initialization (PaI) Mechanisms**
>
> | Method | Saliency Criterion / Mechanism | Data-Dependent? | Pruning Schedule | Mathematical Formulation |
> | :--- | :--- | :--- | :--- | :--- |
> | **YOPO (Ours)** | **Nonnegative low-rank residual magnitude** | **No** | **One-Shot (Zero-Shot)** | **$S = \|\tilde{W} - VH\|$, where $\tilde{W} = \|W\|$ is approximated by NMF factors $V \ge 0, H \ge 0$. Mask $m = \mathbf{1}[S > \tau]$, where $\tau = \text{median}(S) + \alpha \cdot \text{MAD}(S)$**  |
> | **DPaI** (Xiang et al., 2025) | Differentiable Node-Path Balancing (d-NPB) via continuous mask optimization | No | Continuous Optimization (via gradient descent) | $\max_{s} (1-\alpha) \log R_P + \alpha[(1-\beta) \log R_N + \beta \log R_C]$ using Straight-Through Estimator on $m = \text{Top}_k(\|s\|)$ |
> | **NPB** (Pham et al., 2023) | Node-path balancing via discrete optimization | No | Layer-wise discrete optimization | $\max_M \alpha R_N + (1-\alpha)R_P$ s.t. $\|M\|_1 \le N(1-\tau)$ |
> | **PHEW** (Patil & Dovrolis, 2021) | Random walk biased by weight magnitude | No | Iterative | Transition probabilities $P \propto \|w\|$ to preserve input-output paths |
> | **Iter-SNIP** (De Jorge et al., 2020) | Connection sensitivity to training loss | Yes | Iterative | $z = \|w \odot \nabla_w L\|$ computed and pruned gradually |
> | **GraSP** (Wang et al., 2020) | Hessian-gradient product | Yes | One-Shot | $z = -w \odot (H \nabla_w L)$ |
> | **SynFlow** (Tanaka et al., 2020) | Synaptic flow conservation | No | Iterative | $z = \|w \odot \nabla_w R\|$, where $R = \mathbf{1}^\top (\prod \|w_l\|) \mathbf{1}$ |
> | **SNIP** (Lee et al., 2019) | Connection sensitivity to training loss (Gradients) | Yes (1 mini-batch) | One-Shot | $z = \|w \odot \nabla_w L\|$ |
> | **Magnitude** | Absolute initial weight magnitude | No | One-Shot | $z = \|w\|$ |
> | **Random** | Uniform random scoring | No | One-Shot | $z \sim \mathcal{U}(0,1)$ |

---

### Author Response · Authors · 2026-07-07
**Authors' Response: Camera-Ready Revision Summary**

We thank the Action Editor and Reviewers for their time, careful reading, and constructive feedback. All revisions are highlighted in blue in the camera-ready manuscript. Below is a brief summary of changes made.


**1. Unstructured pruning discussion (Introduction, Appendix D).** We added a concise, field-level paragraph (*"Unstructured Pruning: Limitations and Benefits"*) discussing why unstructured PaI cannot directly yield wall-clock speedups on dense GPU/TPU hardware (detailed further in the Limitations section in Appendix D), alongside its practical motivations: finest-granularity parameter selection at initialization, direct model-size reduction, and emergent coarse-grained sparsity patterns as a bridge toward structured compression.

**2. LLM generality (Appendix P).** We added Appendix P, with sufficient experimental details, evaluating YOPO on LLaMA-2-7B and Phi-2 (2.7B) at 20%, 25%, and 30% unstructured sparsity using data-free NMF residuals (rank $r=7$) on MLP weight matrices, with attention layers and layer norms left dense. Across five zero-shot reasoning benchmarks, YOPO retains 99.2% of LLaMA-2-7B dense accuracy at 30% sparsity and matches or exceeds dense accuracy on Phi-2, substantially outperforming SliceGPT and PruneNet at 20% compression. Wall-clock throughput (A100-40GB, batch 8) is also reported to transparently confirm the expected limitation of unstructured sparsity without sparse kernels.

**3. PaI mechanism comparison table (Table 12).** We incorporated a structured comparison table summarising the saliency criterion, data dependence, pruning schedule, and mathematical formulation for all baselines and YOPO. This table is now integrated into the manuscript.

**4. Formatting.** All revisions are highlighted using `\color{blue}` / `revenv`. New bibliography entries for LLM baselines and surveys have been added.


We hope these changes satisfactorily address all comments, and we are grateful for the opportunity to improve the paper.

---

> ### Comment · Action_Editor_Q3Bj · 2026-07-20
> **Please submit camera-ready version**
>
> Thank you for the revision. Please submit the camera-ready version (undo the highlighting).

---

### Decision · Action_Editor_Q3Bj · 2026-06-25

**Recommendation:** Accept with minor revision

**Additional Comments:**

The authors have presented in their replies a number of additional results, but they were not incorporated into a revision. These results should be incorporated into the manuscript with sufficient experimental details.
The limitations (and potential benefits) of unstructured pruning methods should be discussed in the main text (Introduction).
The authors should indicate changes in color to ease comparison with the current version.

**Audience:**

Yes

**Audience Explanation:**

The results are of potential interest to researchers concerned with sparse neural networks and network pruning.

**Claims And Evidence:**

Yes

**Claims Explanation:**

The main claim of the paper is an efficient data- and gradient-free pruning method for neural networks. They provide empirical results for this claim. They perform additional analyses to provide insights into their pruning method.